



# Morphological evolution of bifurcations in tide-influenced deltas

**Arya P. Iwantoro, Maarten van der Vegt, and Maarten G. Kleinhans**

Department of Physical Geography, Faculty of Geosciences, Utrecht University, Utrecht,
3508 TC, the Netherlands

**Correspondence:** Arya P. Iwantoro (a.p.iwantoro@uu.nl)

**Abstract.** In river-dominated deltas, bifurcations often develop an asymmetrical morphology; i.e. one of the downstream channels silts up, while the other becomes the dominant one. In tide-influenced systems, bifurcations are thought to be less asymmetric and both downstream channels of the bifurcation remain open. The main aim of this study is to understand how tides influence the morphological development of bifurcations. By using a depth-averaged (2DH[CE1]) morphodynamic model (Delft3D), we simulated the morphological development of tide-influenced bifurcations on millennial timescales. The schematized bifurcation consists of an upstream channel forced by river discharge and two downstream channels forced by tides. Two different cases were examined. In the first case, the downstream channels started with unequal depth or length but had equal tidal forcing, while in the second case the morphology was initially symmetric but the downstream channels were forced with unequal tides. Furthermore, we studied the sensitivity of results to the relative role of river flow and tides. We find that with increasing influence of tides over river, the morphology of the downstream channels becomes less asymmetric. Increasing tidal influence can be achieved by either reduced river flow with respect to the tidal flow or by asymmetrical tidal forcing of the downstream channels. The main reason for this behaviour is that tidal flows tend to be less unequal than river flows when geometry is asymmetric. For increasing tidal influence, this causes less asymmetric sediment mobility and therefore transport in both downstream channels. Furthermore, our results show that bedload tends to divide less asymmetrically compared to suspended load and confirm the stabilizing effect of lateral bed slopes on morphological evolution as was also found in previous studies. We show that the more tide-dominated systems tend to have a larger ratio of bedload-to-suspended-load transport due to periodic low sediment mobility conditions during a transition between ebb and flood. Our results explain why distributary channel networks on deltas with strong tidal influence are more stable than river-dominated ones.

## 1 Introduction

Deltas often consist of distributary channel networks. In these systems, water and sediment are divided at the bifurcations and distributed over the delta. The shape of the delta and the number of active channels depends on many factors like the forcing by rivers, tides, and waves (Galloway, 1975; Rossi et al., 2016; Shaw and Mohrig, 2014); sediment availability, and sediment type (Geleynse et al., 2011). Bifurcations tend to develop differently in river- than in tide-dominated systems, because tides influence the mouth bar

formation processes of active river-dominated deltas (Edmonds and Slingerland, 2007; Leonardi et al., 2013; Shaw and Mohrig, 2014). In tidal deltas tides propagate upstream and can induce bidirectional flows. This unique characteristic may lead to a different morphological evolution of the bifurcations than would occur in the river-dominated zone (Frings and Kleinhans, 2008; Hoitink et al., 2017), but this has not been proven yet and the underlying mechanisms have not been studied. The focus of this paper is on the stability and depth asymmetry of bifurcations in tidally influenced deltas. We do not focus on the morphological evolution of the

entire delta or the formation process of mouth bars, but we consider a single bifurcation consisting of one upstream and two downstream channels. These are the building blocks of deltas, and the hydro- and morphodynamics of such a system have been studied before by many others (Wang et al., 1995; Bolla Pittaluga et al., 2003; Buschman et al., 2010, 2013; Kleinhans et al., 2008; Sassi et al., 2011).

In river-dominated systems, the morphology of the downstream channels of bifurcations often develops asymmetrically, such that one downstream channel deepens while the other silts up (Kleinhans et al., 2008). In many cases this condition develops into an avulsion. This asymmetric development can be triggered by a small perturbation such as a different bed elevation at the junction (Bolla Pittaluga et al., 2003), by a meandering upstream channel nearby the bifurcation, or by the geometry of the downstream channels such as different lengths of the downstream branches (Kleinhans et al., 2008). The study of this morphological evolution in river-dominated bifurcations was pioneered by Wang et al. (1995). They applied an analytical model to predict the stability of river bifurcations. They found that bifurcations can be stable if any tendency for a downstream branch to become more dominant is counteracted by a relatively large share of the sediment input. Bolla Pittaluga et al. (2003) improved the model in Wang et al. (1995) by taking into account the cross-channel flow that can be induced by an asymmetric cross-sectional profile at the bifurcation. This effect induces a lateral bedload transport, which affects the asymmetric sediment division to the downstream branches. Using this approach, they found that the asymmetry of depth of the two downstream branches depends on the Shields number and on the width-to-depth ratio of the upstream channel at the bifurcation. Bifurcations with high width-to-depth ratio and low Shields number will be unstable and develop asymmetrical depths. Bertoldi and Tubino (2007) confirmed the results by Bolla Pittaluga et al. (2003) using a physical-scale model. Kleinhans et al. (2008) proposed that this asymmetrical depth development is also influenced by meandering of the upstream channel. The meandering bend induces an asymmetrical cross-sectional bed profile and thereby influences the division of sediment at the junction. Bolla Pittaluga et al. (2015) continued the work of Bolla Pittaluga et al. (2003) for a wider range of sediment mobility conditions. They found a range of sediment mobility numbers that result in stable symmetric bifurcations. Meanwhile, bifurcations with sediment mobility higher or lower than this range will grow asymmetrically and avulse. Applying the concept of Bolla Pittaluga et al. (2003), Salter et al. (2017) showed that deposition of sediment at a relatively shallow shelf causes the shorter channel to lengthen and reduce in gradient, thereby balancing the sediment transport division between downstream channels with unequal lengths. Redolfi et al. (2016) eliminated the need for a calibrated parameter in the lateral bedload transport by Bolla Pittaluga et al. (2015), and, by using that approach, Redolfi et al. (2019) showed

that stable, symmetric bifurcations can only occur when the width-to-depth ratio of the upstream channel is below the critical limit originally defined in the theory of meandering rivers by Blondeaux and Seminara (1985), where the critical limit value depends on the friction and Shields stress at bifurcation.

In contrast to our knowledge of morphological development of bifurcations in river-dominated systems, our knowledge of this particular area in tide-influenced systems is still limited. Observations suggest that a similar development as in river-dominated systems can occur, as, for example, found in the most upstream bifurcation of the Yangtze Estuary that divides the main channel into the North Branch and South Branch. According to Chen et al. (1982), the North Branch has evolved to be narrower and shallower, while the South Branch has deepened. However, bifurcations in other tide-influenced deltas have downstream channels that seem to have a less asymmetric depth distribution, e.g. the Berau River delta (Buschman et al., 2013) and Kapuas River delta (Kästner et al., 2017). It has been suggested that tidal deltas have more stable distributary channel networks than their river-dominated counterparts (Hoitink et al., 2017), but the underlying mechanisms are unknown. Furthermore, several studies have investigated tidal characteristics at tidal bifurcations. Despite a general understanding on tides and subtidal water division at tidally influenced bifurcations (Buschman et al., 2010, 2013; Sassi et al., 2011; Zhang et al., 2012; Alebregtse and de Swart, 2016), the effect of tides on the morphological evolution of tidal bifurcations has not been fully understood yet. From previous studies it is clear that tides influence the subtidal flow (Buschman et al., 2010; Sassi et al., 2011) and sediment division (Buschman et al., 2013), induce tidal currents that influence the sediment mobility, and can cause cross-channel currents at the junction (Buschman et al., 2013; Kleinhans et al., 2013). In river systems, all these factors are important for the morphological development of the downstream channels and it is expected that this is also the case for tide-influenced systems.

Therefore, the main aim of this paper is to study the effect of tides on the morphological evolution of bifurcations with the focus on how tides contribute to the asymmetrical development. For this purpose, an idealized bifurcating channel was set-up in Delft3D. We simulated the morphological evolution of a system consisting of two downstream channels (branches) forced by tides and an upstream channel forced by river discharge. We consider this system as a building block of each delta system. We studied two cases, i.e. asymmetric geometry of downstream channels and asymmetric tides between the downstream channels. In the former case, the asymmetric downstream geometry was initially prescribed to see how tides affect the asymmetrical development of the downstream channels. The relative effect of tides was investigated by imposing equal tides at downstream boundary of each downstream branch and by using different values for the river discharge in a series of simulations. In the latter case,

we imposed unequal tidal forcing at the two downstream boundaries that had a symmetric geometry. In tide-influenced deltas, the asymmetric tides between downstream channels can occur because the downstream channels are connected to other channels with different complexity, which may dissipate the tidal range or slow down the tides unequally before the tides propagate into the downstream channels of the bifurcation.

This paper is organized as follows. The model set-up and methodology are described in Sect. 2. In Sect. 3, the results of the simulated morphological development are presented. Section 4 presents a discussion on the findings. Finally, the conclusions of this study are provided in Sect. 5.

## 2 Methodology

### 2.1 Model set-up

An idealized bifurcating channel was set up and its morphological development was simulated using the depth-averaged version (2DH) of Delft3D. This 2D approach is suitable for long-term large-scale morphodynamic modelling, because it is computationally lighter than a 3D approach. Even though a 3D approach allows for vertical flow patterns (Lane et al., 1999) such as curvature-induced flow, which might be important for the sediment transport process (Daniel et al., 1999), the 2D approach is sufficient for this study since we focus on large-scale morphodynamic evolution and therefore simulating detailed 3D features of flow and morphology is not our goal. Furthermore, the reason to prefer the 2D above the 1D approach is to explicitly simulate cross-channel flow induced by tidal propagation from one branch to another at the junction as observed in Buschman et al. (2010, 2013) and as being identified by Bolla Pittaluga et al. (2003) as an important process for sediment division at the junction.

The model solved the 2DH unsteady shallow water equations using a semi-implicit alternating direction implicit (ADI) scheme on a staggered grid (see Lesser et al., 2004). For bed friction, the Chézy formulation was used with a value of $60 \, \mathrm{m}^{1/2} \, \mathrm{s}^{-1}$. Meanwhile the horizontal eddy viscosity was set to $10 \, \mathrm{m}^2 \, \mathrm{s}^{-1}$. This value was chosen because applying a smaller value for horizontal eddy viscosity will cause a numerical instability near the bifurcation as flow magnitude and direction rapidly change in this location and applying a larger value will not significantly affect the results. Bedload and suspended load sediment transport were calculated by the van Rijn (1993) method. We used medium sand with a single grain size of 0.25 mm with a dry bed density of $1600 \, \mathrm{kg} \, \mathrm{m}^{-3}$. This sediment size is in the range of observed grain size in tide-influenced deltas, as, for example, by Buschman et al. (2013) in the Berau River delta (0.125–0.25 mm), Kästner et al. (2017) in the Kapuas Delta (0.22–0.3 mm), Sassi et al. (2011) in the Mahakam Delta (0.25–0.4 mm), and Stephens et al. (2017) in the Mekong Delta (0.074–0.385 mm). Transverse bed-slope effects for

bedload transport were accounted for by the approach of Ikeda (1982), and we used a value of 10 for $\alpha_{\mathrm{bn}}$. This value is much higher than the Delft3D default value (1.5) and that suggested by Bolla Pittaluga et al. (2003) (0.3–1), because a low value of this parameter in Delft3D leads to unrealistic and grid-size-dependent channel incision as well as bar formations (Baar et al., 2019). Even though we prescribed a high $\alpha_{\mathrm{bn}}$, this value is still in the range of what other studies used for Delft3D modelling work (e.g. Dissanayake et al., 2009; van der Wegen and Roelvink, 2008, 2012). For streamwise bed-slope effects, the Bagnold (1966) approach was used with a Delft3D default value of $\alpha_{\mathrm{bs}} = 1$. For morphology, the MorFac approach was used (Lesser et al., 2004; Roelvink, 2006) with an acceleration factor of 400. We tested several values between 1 and 1000, and we chose the largest value for which morphology had similar development as for a value of 1 and numerical stability was satisfied. This allows for long-term morphodynamic simulation at timescales of decades (Lesser et al., 2004) and centuries (TS1 van der Wegen et al., 2008) in a much shorter duration. Furthermore, in this study, non-erodible channel banks were used. This limitation was acceptable since changes in width-to-depth ratio could still be accommodated by the bed level change and using erodible banks is not realistic as long as the model is not able to allow for channel bank growth.

The spatial domain consisted of an upstream channel that bifurcates in two downstream channels. The two downstream channels had a default length of 30 km; although, in one series of simulations the length of one channel was 15 km. The upstream channel had a length of 220 km to ensure that upstream propagating tides decay smoothly. The downstream channels and the first 20 km of the upstream channel had a convergent width profile, while the upstream 200 km had a constant width. The channel width was configured by TS2

$$W_{\mathrm{upstream}}(x) = \begin{cases} W_0 e^{-x/L_{\mathrm{w}}}, & \text{for } x < 20 \, \mathrm{km}, \\ W_0, & \text{for } x > 20 \, \mathrm{km}, \end{cases}$$

$$W_{\mathrm{downstream}}(x) = 0.5 W_0 e^{-x/L_{\mathrm{w}}}, \tag{1}$$

in which $W(x)$ is the channel width, $x$ the longitudinal distance from the junction (i.e. positive in TS3 upstream direction, $x = 0$ at bifurcation, and hence $x$ is negative in downstream channels), $W_0 = 322 \, \mathrm{m}$ is the width at the junction and $L_w = 50 \, \mathrm{km}$ is the $e$-folding length scale. Further, in a region within 800 m near the junction, an additional widening was applied (Fig. 1b) to overcome the loss of two grid cells (see grid description in Kleinhans et al., 2008). This widening is a typical feature of bifurcations found in delta systems (Kleinhans et al., 2008). After the additional widening, $W_0$ becomes 750 m.

The spatial domain of the model was discretized in a curvilinear grid and followed the same method as in Kleinhans et al. (2008) and Buschman et al. (2010). At the bifurcation two grid cells had to be removed in the middle of the channel for numerical reasons (Kleinhans et al., 2008), as illustrated in

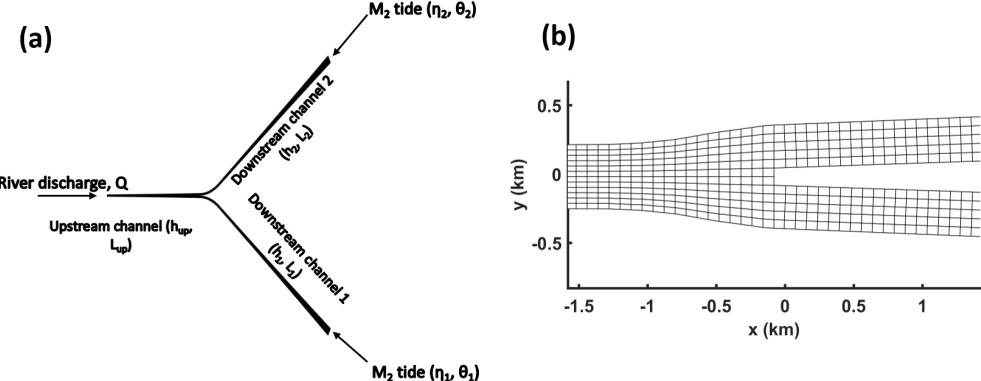

**Figure 1. (a)** Illustration of bifurcation model set-up from the upstream channel forced by river discharge $Q$ to the downstream boundaries which are forced by tidal water levels. Here, $h$ indicates the depth and $L$ indicates the length of each channel. Meanwhile, $\eta$ and $\theta$ indicate amplitude and phase of tidal water levels at each downstream boundary. **(b)** Zoom of model grid near the junction, showing the additional widening near the junction and the disappearance of two grid cells downstream of the junction.

Fig. 1. The grid cell length in the along-channel direction was 80 m. The upstream channel had 12 grid cells across the channel, whereas in both downstream channels 5 grid cells were used. Therefore, the grid cell size in the across-channel direction was spatially varying in order to adapt the funnelling shape of the channel and the additional widening near the bifurcation. Near the junction this resulted in a typical grid cell width of 40 m. Based on grid size and channel depth, a time step of 6 s was used in all simulations to have a Courant number smaller than $4\sqrt{2}$ as required for the ADI scheme. The domain had three open boundaries where boundary conditions for flow and sediment transport were prescribed. At the upstream end of the upstream channel river discharge was prescribed, while at the ends of the two downstream channels $M_2$ tidal water levels were imposed. At all open boundaries, equilibrium sediment transport was computed during inflow, while during outflow the sediment transport was assumed to be just flushed out from the domain. As a result, no morphological change occurred during inflow, but the bed is free to evolve during outflow.

Because the formation of alternating bars will affect flow and sediment division at the junction, the channel depth and upstream-prescribed river discharge were chosen such that the system was in the overdamped bar regime (Struiksma et al., 1985). To this end, we conservatively followed the empirical classification proposed by Kleinhans and van den Berg (2011). Therefore, the three connected channels had an initial depth of 15 m and a constant along-channel bed slope of $3 \times 10^{-5}$ m m$^{-1}$. The prescribed discharge ranged between 500 and 2800 m$^3$ s$^{-1}$.

## 2.2 Description of model scenarios and boundary conditions

Depth, width, and length of the downstream channels of bifurcations in deltas can be unequal. Hence, in Case 1 we

started the simulations with an unequal geometry, either being a difference in depth or length between the two downstream channels. We simulated the morphological evolution of the bifurcation until it approximately reached morphodynamic equilibrium (discussed later on). Note that the length of the branches was fixed in time, while an initial depth difference does not necessarily result in an asymmetric equilibrium depth because it can adapt. All simulations belonging to Case 1 were forced by equal tides from downstream and river discharge from upstream (settings summarized in Table 1). The depth difference scenarios were performed in two different ways. First, simulations were started from a system in which the upstream channel and one downstream channel were 15 m deep, while the other branch was 7.5 m deep (called Depth1). The upstream 2 km of the shallow downstream channel was gradually changed over 2 km to avoid a sudden depth change near the bifurcation. In a second type of simulation, we started with uniform bathymetry of 15 m depth and simulated until morphodynamic equilibrium was reached (called Depth2). Next, one downstream channel was made 0.5 m deeper and the other 0.5 m shallower. We studied the sensitivity of the results to the relative magnitude of tides over river discharge by changing the prescribed upstream discharge. The simulation with largest river discharge (2800 m$^3$ s$^{-1}$) represents a river-dominated system, while the simulations with lower river discharge (500 m$^3$ s$^{-1}$) represent the more tide-influenced systems.

In Case 2 the effect of unequal tidal forcing on morphological development was studied. In natural systems tides in the two downstream branches can be unequally forced. For example, when the two branches end in a shelf sea, amplitude and phase in the two channels can be different because they have a different position with respect to the amphidromic system in the shelf sea. Furthermore, in deltas with multiple bifurcations and unequal depths and channel lengths, tidal amplitude and phase differences will be present in the chan-

**Table 1.** Summary of simulations undertaken in the present study and their boundary conditions (river discharge and tidal properties), as well as geometry differences between the downstream channels.

| Scenario | Simulation name | $Q$ (m$^3$ s$^{-1}$) | $\eta_{M_2}$ (m) | | $\Delta\theta_{M_2}$ (°) | $\Delta L$ (km) | $\Delta h$ (m) |
|---|---|---|---|---|---|---|---|
| | | | Branch 1 | Branch 2 | | | |
| Control simulation | Control_Q2800 | 2800 | 1 | 1 | 0 | 0 | 0 |
| | Control_Q1596 | 1596 | 1 | 1 | 0 | 0 | 0 |
| Depth difference | Depth1_Q2800 | 2800 | 1 | 1 | 0 | 0 | 7.5 |
| | Depth1_Q1596 | 1596 | 1 | 1 | 0 | 0 | 7.5 |
| | Depth1_Q500 | 500 | 1 | 1 | 0 | 0 | 7.5 |
| | Depth2_Q2800 | 2800 | 1 | 1 | 0 | 0 | 1 |
| | Depth2_Q1596 | 1596 | 1 | 1 | 0 | 0 | 1 |
| Length difference | Length_Q2800 | 2800 | 1 | 1 | 0 | 15 | 0 |
| | Length_Q1596 | 1596 | 1 | 1 | 0 | 15 | 0 |
| Amplitude difference | Amp_0.75 | 2800 | 0.75 | 1 | 0 | 0 | 0 |
| | Amp_0.5 | | 0.5 | 1 | 0 | 0 | 0 |
| | Amp_0.25 | | 0.25 | 1 | 0 | 0 | 0 |
| Phase difference | Phase_10 | 2800 | 1 | 1 | 10 | 0 | 0 |
| | Phase_22.5 | | 1 | 1 | 22.5 | 0 | 0 |
| | Phase_35 | | 1 | 1 | 35 | 0 | 0 |

nels because propagation speeds and times in the channels are different. Hence, in Case 2 we started simulations with a symmetric geometry but with asymmetric tidal forcing, either being a tidal water level amplitude difference or a tidal phase difference. The corresponding settings of the simulations can be found in Table 1. The difference in downstream tidal forcing between the two channels was studied for values between 0 and 0.75 m ($\eta_1$ in Fig. 1 was 0.75, 0.5, or 0.25 m, while $\eta_2$ was 1 m) where $\eta_1$ and $\eta_2$ are tidal water level amplitude imposed at the downstream end of downstream channels 1 and 2, respectively. Meanwhile for another set of simulations, the tides had equal amplitude but the phase difference was 10, 22.5, or 35° (for $M_2$ tide this means one channel had delayed tides of 20, 46, or 72 min).

We also performed two control simulations with different discharge, symmetric geometry, and equal tides (see Table 1) to study the equilibrium bed profiles in the absence of any initial asymmetry. The morphology change simulated for Case 1 and Case 2 were caused by the asymmetric forcing/geometry and by the adaptation to the initial conditions. Therefore, the results of the control simulations can be used to better interpret the simulations of Case 1 and Case 2.

## 2.3 Methods to evaluate model simulations

The morphological development of the bifurcation was observed by evaluating for each downstream channel the tidally and spatially averaged depth of the first 2 km from the bifurcation (Fig. 2, called $h_1$ and $h_2$ hereafter). This region was chosen because it determined the morphological development of the entire downstream channel. The development

of the downstream channels starts from upstream and develops downstream. Therefore, analysing the most upstream end of the downstream channels is sufficient to determine the growth in asymmetry between them. After analysing all cases, it was found that a distance shorter than 2 km cannot be representative due to the presence of local morphological features near the bifurcation such as bar formation or small incisions in the downstream channel that is silting up. However, a longer distance cannot be representative because even though one downstream channel almost avulsed upstream, tides can cause a deepening of that same channel near the downstream boundary. To determine whether the system was in morphodynamic equilibrium, we analysed the evolution in time of $h_1$ and $h_2$. We stopped the simulation when the changes in $h_1$ and $h_2$ were small. A true morphodynamic equilibrium, in the sense that no bed level change occurred in the entire domain, was never achieved. This is very common for morphodynamic simulations of estuaries (Van Der Wegen and Roelvink, 2008; Nnafie et al., 2018). A typical simulated period was between 1200 and 2400 years, depending on the prescribed river discharge.

To compare the depth of the two downstream channels, the depth asymmetry parameter $\Psi_h$ was calculated as

$$\Psi_h = \frac{|h_2 - h_1|}{h_1 + h_2}. \tag{2}$$

A larger $\Psi_h$ indicates a more asymmetric morphology. When $\Psi_h$ is close to one, this indicates an avulsion, given that the widths are fixed, while a zero value indicates equal depth of the downstream channels.

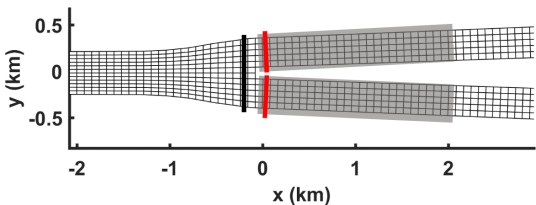

**Figure 2.** The grids in the surroundings of the bifurcation overlaid by the areas where the bed level changes were evaluated (grey boxes) and the grids where the asymmetry indices (red lines) and upstream channel flow (black line) were calculated.

The sediment mobility was evaluated by calculating the width-averaged value of the Shields number two grid cells away from the bifurcation, as illustrated in Fig. 2. The Shields number at each grid point was calculated as

$$\tau_* = \frac{\tau_b}{(\rho_s - \rho_w)\, g\, D_{50}}, \tag{3}$$

where $\rho_s - \rho_w = 1650\,\mathrm{kg\,m^{-3}}$, $g$ is gravitational acceleration $(9.81\,\mathrm{m^2\,s^{-1}})$, and $\tau_b$ is the bed shear stress magnitude expressed by

$$\tau_b = \frac{\rho_w\, g\, u^2}{C^2}, \tag{4}$$

in which $C$ is the Chézy coefficient and $u$ is instantaneous flow velocity. In tide-influenced systems, tides cause a temporal change of bed shear stress, and we calculated both the peak and the tide-averaged value of the Shields number. A Shields asymmetry parameter $\Psi_{\tau_*}$ was defined and calculated by TS4

$$\Psi_{\tau_*} = \frac{\left|\Delta|\overline{\tau_{*;1,2}}|\right|}{|\overline{\tau_{*,1}}| + |\overline{\tau_{*,2}}|}, \tag{5}$$

where $|\overline{\tau_{*,1}}|$ and $|\overline{\tau_{*,2}}|$ are the width-averaged Shields number in each downstream channel, and $\Delta|\overline{\tau_{*;1,2}}|$ is the difference between both. A higher value of $\Psi_{\tau_*}$ indicates a more asymmetric sediment mobility condition, while $\Psi_{\tau_*} = 0$ indicates a symmetric sediment mobility. When $\Psi_{\tau_*}$ was based on peak bed shear stresses, it is denoted by $\Psi_{\tau_* \mathrm{max}}$ TS5, while $\Psi_{<\tau_*>}$ is used when it is based on tidally averaged bed shear stresses.

At the grid locations where we determined the Shields number, we also determined the tidally averaged $(U_0)$ and the $M_2$ tidal $(U_{M_2})$ flow magnitudes, in a similar way as for the Shields number. Furthermore, we calculated the width-integrated and tidally averaged bedload and suspended load transport at the cross sections shown in Fig. 2.

## 3   Results

### 3.1   Evolution of control runs

Results of the two control simulations show that bed levels were initially not in morphodynamic equilibrium. The time-stack diagram of width-averaged depth as a function of space is shown in Fig. 3. The morphology changed over time until an approximate equilibrium was reached, which took about 1200 years. There are two timescales involved. First, there are deposition fronts from the upstream channel that migrate downstream. Second, there is a slower adaptation to the equilibrium condition. The results also show that true morphodynamic equilibrium, in the sense that bed levels are steady, was not achieved after 1200 years. However, bed level changes were small at the end of the simulation. The lowest discharge resulted in the smallest depth for the upstream channel, but the river discharge does not significantly affect the depth of the two downstream channels. This is because both control simulations were imposed by the same tidal forcing, and the morphology of the downstream channels is mainly controlled by the tides. Typical depths are around 8–10 m for the downstream channels and 10–12 m for the upstream one.

### 3.2   Geometry difference case

When simulations started with unequal channel depth, a similar evolution as the control simulations occurred. The morphological evolution was characterized by three typical timescales. First, there was erosion near the bifurcation, mainly because of the decrease in the cross-sectional area directly seaward of the bifurcation. Second, this erosion was followed by deposition fronts that migrated downstream during the simulation. This deposition front can be identified by a rapid decrease of the depth in the downstream channels at the beginning or halfway through the simulation (Fig. 4). It is similar to the evolution of Control_Q2800 and Control_Q1596, but this depositional front was not necessarily similar in the two downstream channels because of the imposed differences in the initial bed level. Furthermore, in the lowest discharge simulations $(Q = 500\,\mathrm{m^3\,s^{-1}})$ it takes much longer for the deposition front to reach the downstream boundary; therefore, it takes much longer before the system is in the steady state. Third, after the initial adaptation phase, the morphology of the channels started to change gradually. Some simulations took 2400 years $(Q = 500\,\mathrm{m^3\,s^{-1}})$ until the morphological changes near the junction were small. Furthermore, the results show that at the end of the simulation the depth of the shallow branch depends on the discharge (Fig. 4). The higher the discharge, the shallower the shallowest branch is. For the deepest branch, it is the other way around. The deepest branch is shallowest for the lowest discharge.

The simulations that were based on perturbed equilibrium depth (Depth2) had a different morphological evolution and final equilibrium than the ones that started with 7.5 m depth difference (results not shown). The Depth2 simulation did not show the fast, initial depth response, but was mainly characterized by a slow adaptation to a new equilibrium, because the system was still close to equilibrium at the start of the simulation. It took relatively long to achieve the new equilib-

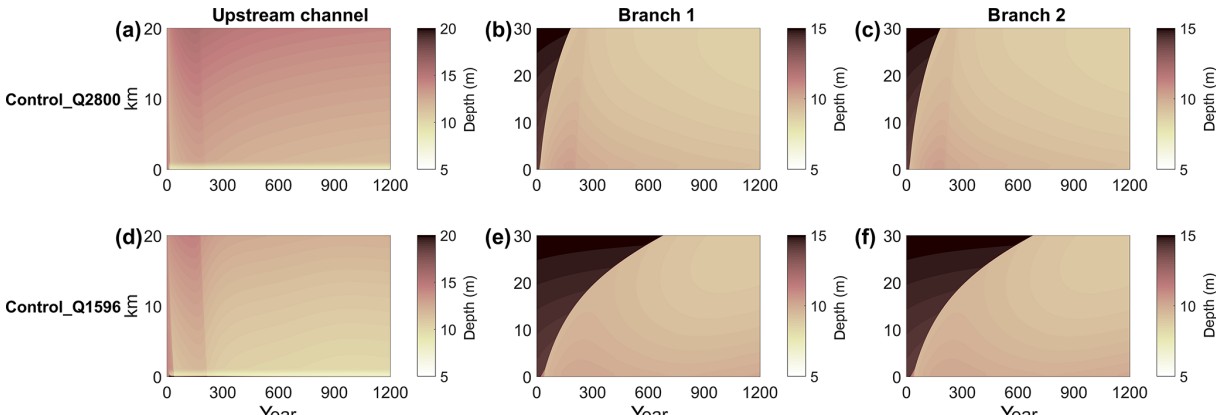

**Figure 3.** Time-stack diagram of width- and tide-averaged depth (colour) of the upstream channel (**a, d**; 0 km is junction, 20 km upstream) and downstream channels (**b, c, e, f**; 0 km is junction, 30 km near sea) as a function of distance from the bifurcation (vertical axis) for the two control simulations. Panels (**a, b, c**) are the result from the high-discharge simulation (Control_Q2800), while panels (**d, e, f**) are for the low-discharge simulation (Control_Q1596).

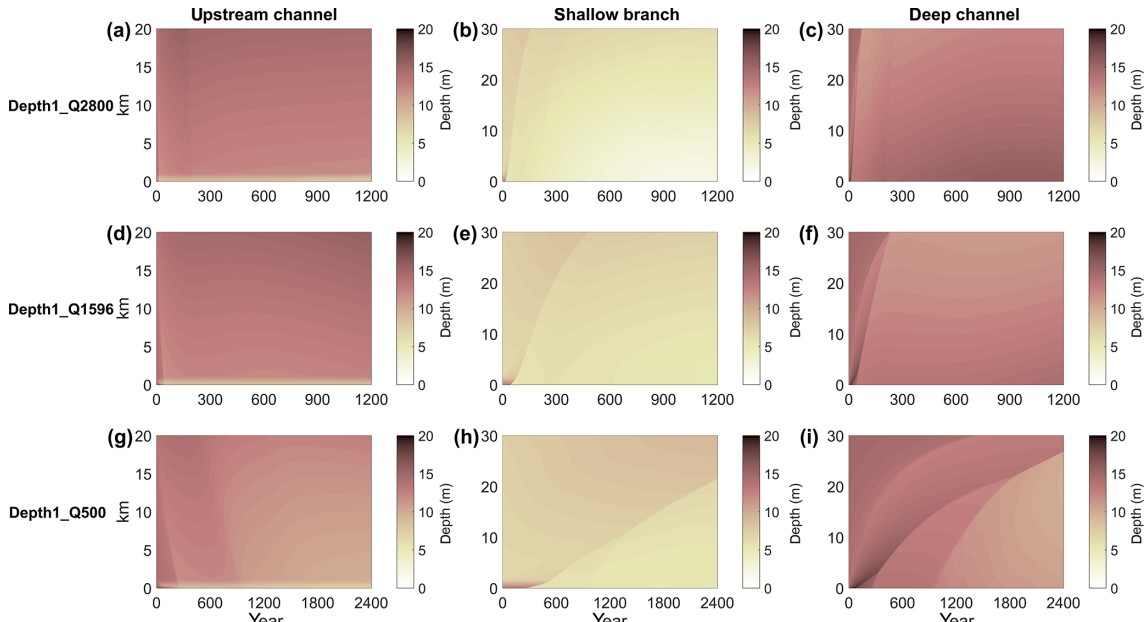

**Figure 4.** Same plot as Fig. 3 but for simulations of Depth1. The panels from top to bottom show the results from different simulations (Depth1_Q2800, Depth1_Q1596, Depth1_Q500, respectively), while from left to right panels show the upstream channel, shallow branch, and deep branch, respectively.

rium and total simulation time was 2400 years in this case. Interestingly, although the external forcing for the Depth1 and Depth2 simulations were the same, the final equilibria were different. Because the depth in the channels influences the tidal dynamics (by, for example, the relative importance of friction and by the difference in tidal propagation speed due to the different initial depths), the tide-induced flows were different at the junction and stayed different during the entire simulation. Hence, the equilibrium not only depends on external forcing but also on initial conditions. The initial

and final morphology near the bifurcation for all Depth1 and Depth2 simulations can be seen in Appendix A.

The simulations with the *length difference* scenario show that the shortest branch developed to be the deepest, while the longest became very shallow (Fig. 5). The longest branch becomes so shallow that it becomes morphologically inactive. This occurred for both the highest and for the medium-discharge scenario, and it is also independent of the initial conditions (starting with equilibrium bathymetry and short-ened channel or with 15 m deep channels). Meanwhile, the shortest channel was deepest for the highest discharge condi-

tion. The final morphology near the bifurcation for the simulations in the *length difference* scenario is provided in Appendix A.

## 3.3 Tide difference case

Asymmetric forcing of tides resulted in asymmetric morphological evolution. Because the system started out of equilibrium, the morphological evolution is again characterized by a quick adaptation followed by a slow evolution to the equilibrium. When forced by different tidal amplitude, the downstream branch with the smallest downstream tidal forcing evolved into the shallowest branch (Fig. 6). Interestingly, when tidal amplitude in Branch 1 was decreased from 0.75 to 0.5 m or even 0.25 m the bifurcation evolved into a less asymmetric system. Furthermore, when the two downstream channels were forced by equal amplitudes, but with different phase, this also resulted in the development of an asymmetric morphology of the bifurcation (Fig. 7). In general, the channel with delayed tides developed smallest channel depth, while the channel with earlier tides developed deeper channels. Interestingly, the deposition front in the shallowest branch became stagnant for the largest imposed phase differences, suggesting that the flow magnitude was below the threshold for erosion (static equilibrium). However, the depth around the bifurcation did not become zero and still evolved. The larger the difference in tidal phase at the two downstream boundaries, the shallower the delayed branch became, while the other branch was deeper. The final morphology near the bifurcation for all simulations of this case is provided in Appendix A.

## 4 Discussion

### 4.1 Relation between tides and the morphological evolution of bifurcations

The results suggest that tides cause less asymmetric bifurcations. To quantify how tides affect the morphology, the results from all scenarios were correlated. Figure 8 shows a scatter plot and linear fit between the final $\Psi_h$ (dimensionless depth asymmetry) and $\Psi_{\tau_*}$ (dimensionless Shields asymmetry) for all model simulations. As can be seen, $\Psi_h$ is linearly correlated with $\Psi_{<\tau_*>}$ and $\Psi_{\tau_*\mathrm{max}}$. Hence, the degree of asymmetry in the morphology is directly related to the degree of asymmetry in the sediment transport capacity. From the comparison of $\Psi_{<\tau_*>}$ and $\Psi_{\tau_*\mathrm{max}}$ against $\Psi_h$ (Fig. 8a and Fig. 8b), the latter comparison shows the strongest relation and therefore the maximum mobility, which occurred during the peak ebb flow in our simulations and is the most representative to determine the morphological asymmetry of the downstream channels.

According to Eq. (3), in a system with uniform sediment properties and water density, the sediment mobility in the downstream channels only depends on the total bed shear

stress $\tau_b$. Because in the downstream channels the flows are mainly in the along-channel direction, the instantaneous flow velocity to calculate the total bed shear stress $\tau_b$ in Eq. (4) can be represented by the along-channel flow velocity. Based on a harmonic analysis, it became clear that the mean flow ($U_0$) and $M_2$ component ($U_{M_2}$) were the main tidal constituents and higher harmonics like $M_4$ were relatively small. Therefore, the maximum sediment mobility scales very well with the square of summation of $U_{M_2}$ and $U_0$ ($\tau_{*\mathrm{max}} \sim (U_{M_2} + U_0)^2$). The sediment mobility and flow conditions near the bifurcation for all simulations is provided in Table A1 in Appendix A.

The relatively strong river discharge in the simulations performed causes the ratios of $U_0$ to $U_{M_2}$ in the upstream channel cross section near the junction to be in the range between 0.2 and values slightly larger than 1 (see Fig. A3a in Appendix A). This similar importance between those components indicates that our model is a mixed river-influenced and tide-influenced system. For most simulations, $U_0$, dominated by the river flow, in the two downstream branches was more asymmetric than $U_{M_2}$ (see Fig. A3b). In river-dominated systems, bifurcations with higher flow division asymmetry will also develop a more asymmetric morphology (Kleinhans et al., 2008). Interestingly, the tidal flows oppose the asymmetry induced by $U_0$. $U_{M_2}$ becomes less asymmetric with the increase of tidal influence, shown by the decreasing trend in $\Psi_{U_{M_2}}$ for increasing sum of $U_{M_2}$ ($\sum U_{M_2}$) in the two downstream channels (Fig. 9a), in which the summed $U_{M_2}$ was measured from the width-averaged $U_{M_2}$ at cross section in the downstream channels shown in Fig. 2. This explains why the increased tidal influence, indicated by larger sum of $U_{M_2}$ in Fig. 9a and b, causes less asymmetric bifurcations. Due to tides, the sediment mobility in both channels is closer to each other than without tides (Fig. 9b). A more tide-influenced condition is not only achieved by decreasing river discharge but also by inducing an asymmetry in the tidal forcing in the downstream channels. For increased difference in either amplitude or phase, the sum of $U_{M_2}$ in both downstream channels also increased and became similar in magnitude. The scatter in the results shown in Fig. 9 is caused by the different imposed asymmetries for different scenarios. The asymmetry is not only controlled by external forcing but also determined by internal dynamics when the depth of the branches develop, and, because we have different types of initial asymmetries (forcing, depth, length), there is quite some scatter in Fig. 9. Still, we found that all simulations have a similar behaviour; i.e. a more tidal influence drives less morphological asymmetry between downstream channels.

There are two processes that drive a less asymmetric tidal flow in the more tide-influenced condition. First, the propagation of tides from the dominant downstream channel to the other downstream channel balances the tidal flow in the two downstream channels. This process mainly rules in the tide difference case. Tidal forcing asymmetry between downstream channels drives tidal propagation from one down-

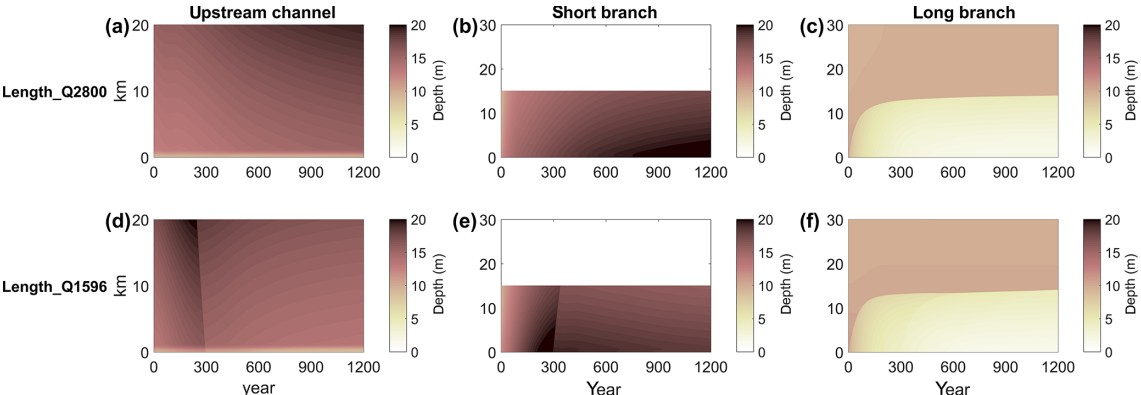

**Figure 5.** Time-stack diagram of width- and tide-averaged depth as a function of space for the simulations in *length difference* scenario with the same order as Fig. 4 but with short (**b, e**) and long (**c, f**) downstream branches.

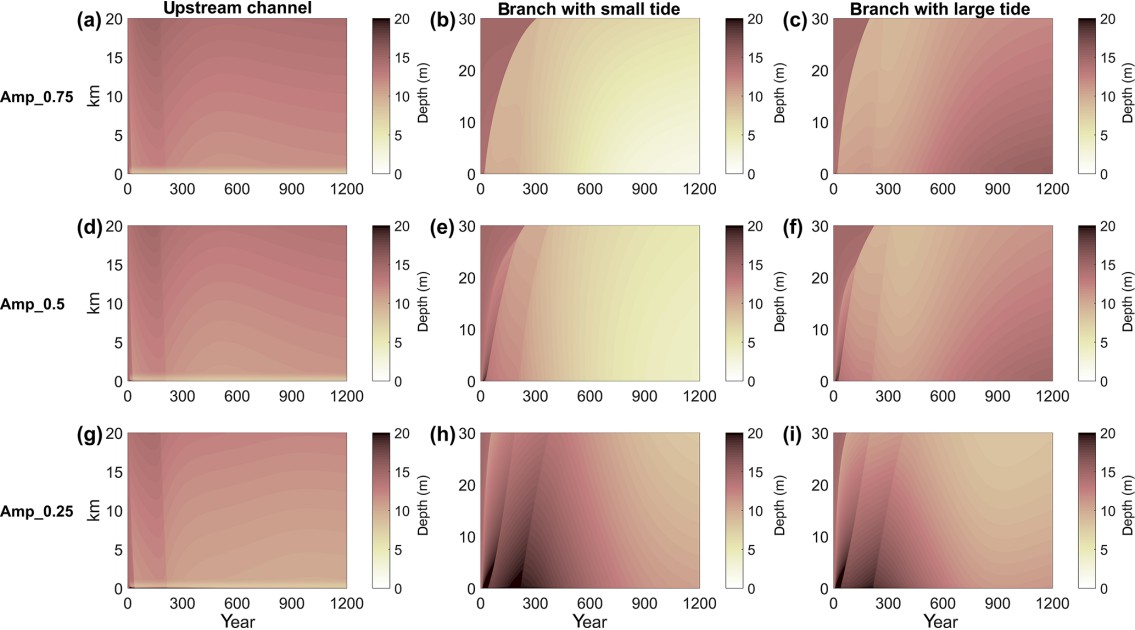

**Figure 6.** Time-stack diagram of width- and tide-averaged depth as a function of space for the *amplitude difference* scenario.

stream channel to the other and results in phase lags of tidal flow inducing strong cross-channel flow at the junction (similarly as discussed in Buschman et al., 2013). This can be seen by a larger cross-channel flow in the upstream channel near the bifurcations for larger asymmetry between the prescribed tides in Fig. 10. This cross-channel flow is dominated by the tides ($V_{M_2}$), while its mean flow value ($V_0$) was close to zero. Strong cross-channel flows caused erosion at the bifurcation, resulting in a trench-like scour connecting the downstream channels. This scour can be found in the *amplitude difference* and *phase difference* scenarios and is most pronounced in the simulation Amp_0.25 (see Fig. A2c–f in Appendix A). Although a bar developed in the upstream channel on the side of the downstream channel imposed with lower tidal amplitude, the cross-channel flows deepened the bed at bi-

furcation and maintained the connection between both downstream channels and the upstream channel. The development of the trench-like scour at the bifurcations is also observed in the Berau River delta (Buschman et al., 2013) and Mahakam Delta (Sassi et al., 2011). This deepening at the bifurcation can be also affected by the angle of the bifurcation (something we did not study here). Second, with equal tides imposed in the two downstream channels for *depth difference* and *length difference* scenarios, the larger river discharge in the dominant downstream channel dampens the tides in this channel, while the shallowing bed level in the other downstream channel increases the tidal flow in this channel. As a result, this combining effect induces a less asymmetric tidal flow in the downstream channels.

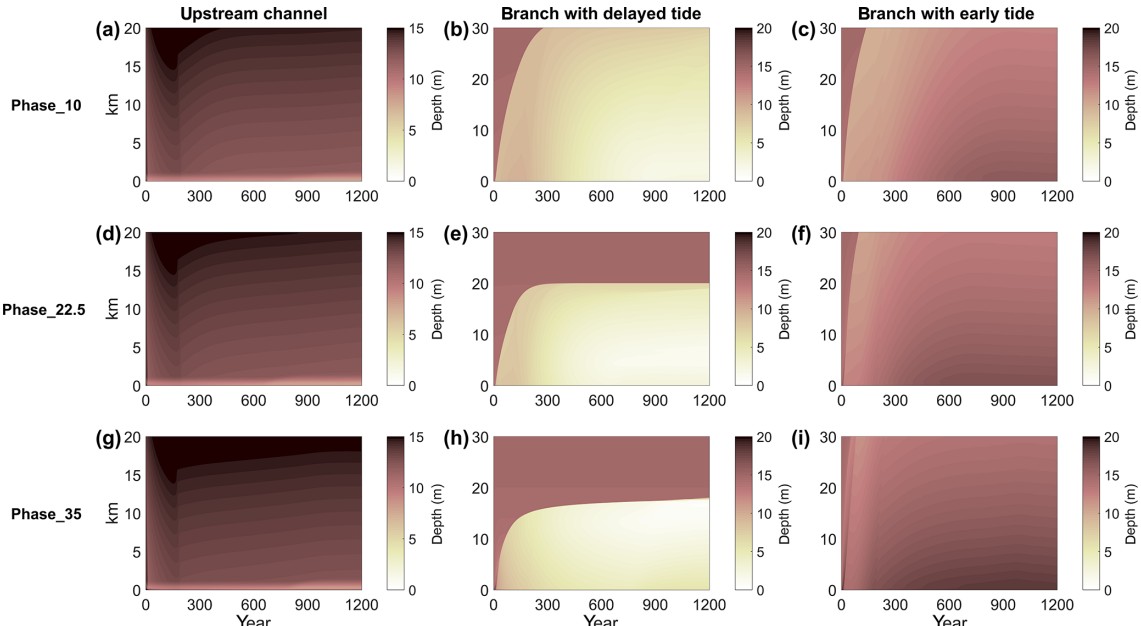

**Figure 7.** Time-stack diagram of width- and tide-averaged depth as a function of space for the *phase difference* scenario.

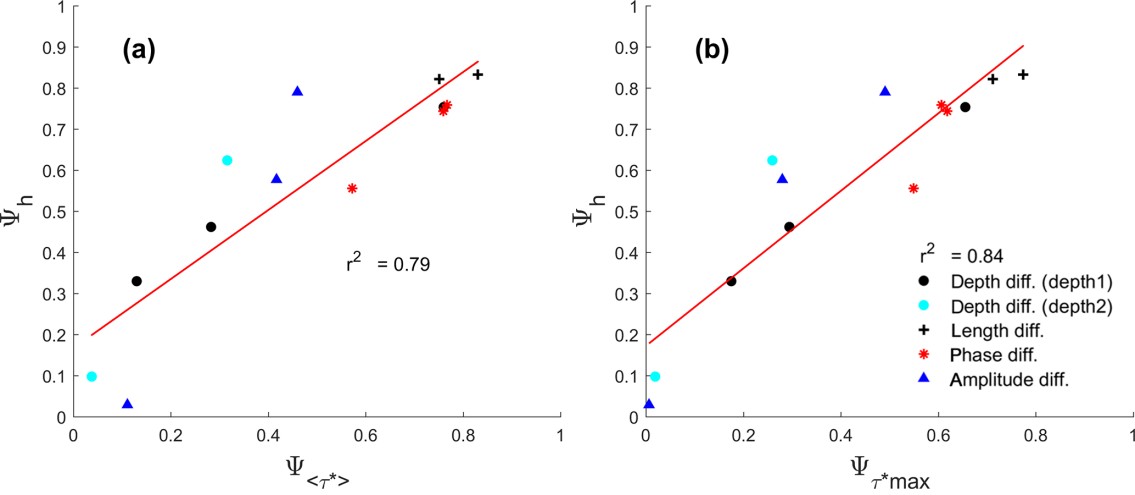

**Figure 8.** Relation between depth asymmetry number $\Psi_h$ and **(a)** the asymmetry in tidally averaged Shields number ($\Psi_{<\tau_*>}$) and **(b)** the asymmetry in peak Shields number ($\Psi_{\tau_* max}$) at the equilibrium condition for all simulations from scenarios described in Table 1. Note that in panel **(a)**, two simulations of the *phase difference* scenario and a simulation of the Depth1 scenario are slightly overlapping.

## 4.2  Role of bedload versus suspended load

In the theory of Bolla Pittaluga et al. (2003, 2015) the lateral bed slope causes additional sediment transport into the dominant channel, thereby having a stabilizing effect on the bifurcation. Here, we used the van Rijn (1993) sediment transport formulations in which bed slope only affects the bedload transport and not the suspended load transport. Based on this, we expected that bedload transport will be divided less asymmetrically than suspended load transport. To check this hypothesis, the tidally averaged and width-integrated sediment

transport at the cross sections shown in Fig. 2 were calculated. We calculated an asymmetry index in a similar way as we did for the Shields number and depth. The results of the scatter plot of suspended load asymmetry versus bedload asymmetry index clearly show that suspended load tends to be divided more asymmetrically at the bifurcation (Fig. 11a). Only when the system is fully symmetric or asymmetric is there no difference in asymmetry of bedload and suspended load transport, because the downstream channels receive an equal amount of sediment when the downstream channels are symmetric, while only one downstream channel receives all

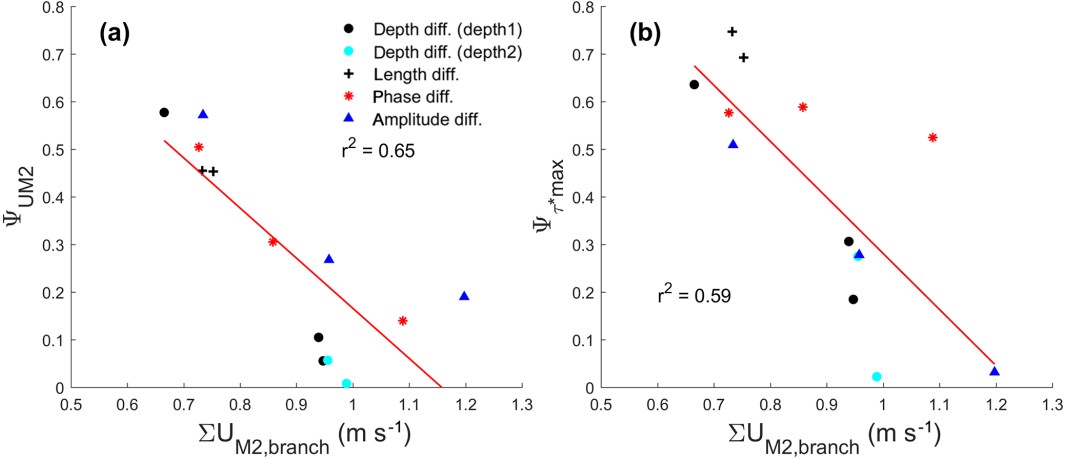

**Figure 9.** Comparison between **(a)** tidal flow asymmetry and **(b)** peak Shields number asymmetry in the two downstream branches against the total tidal flow magnitude from the two downstream channels.

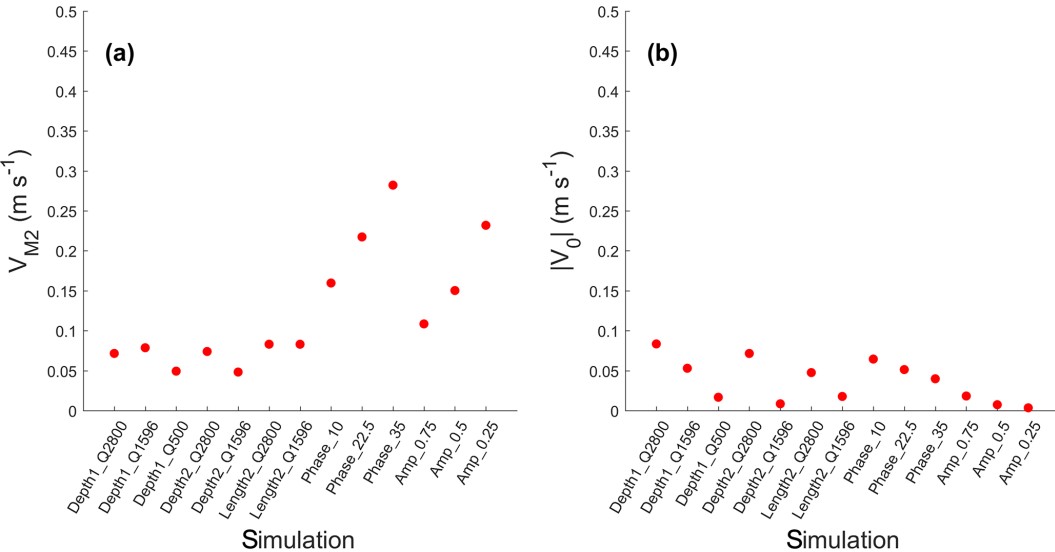

**Figure 10.** Cross-channel flow of **(a)** tidal current amplitude and **(b)** mean current at bifurcation in the upstream channel for all simulations.

sediment when an avulsion occurs (both bedload and suspended load asymmetry are 1). Furthermore, from a scatter plot of depth asymmetry ($\Psi_h$) versus the ratio of bedload to suspended load transport in the upstream channel, it becomes clear that systems that have more asymmetric bed levels have a smaller contribution of bedload transport to the total transport and vice versa (Fig. 11b). However, there is also some considerable scatter due to the sensitivity to the initially imposed asymmetry. Lastly, a scatter plot of the ratio of mean flow and $M_2$ flow magnitude versus the ratio of bedload and suspended load transport in the upstream channel (Fig. 11c) suggest that when river flow is relatively important, the system is dominated by suspended load, while for more tide-dominated conditions bedload plays a more important role.

This further explains why the more tide-dominated conditions result in less asymmetric morphology.

### 4.3 Sensitivity to sediment grain size and lateral bed-slope effect

Defining a different sediment grain size would change the sediment mobility and drive a different ratio of bedload to suspended load transport. These would affect the sediment transport division and therefore the morphological development in the downstream channels. When using finer sediment, this results in a more asymmetric development of the downstream channels, as is shown in Fig. 12. The finer sand induces a larger contribution of suspended load transport to total sediment transport and therefore counteracts the stabilizing effect by the transverse bed-slope effect on the bed-

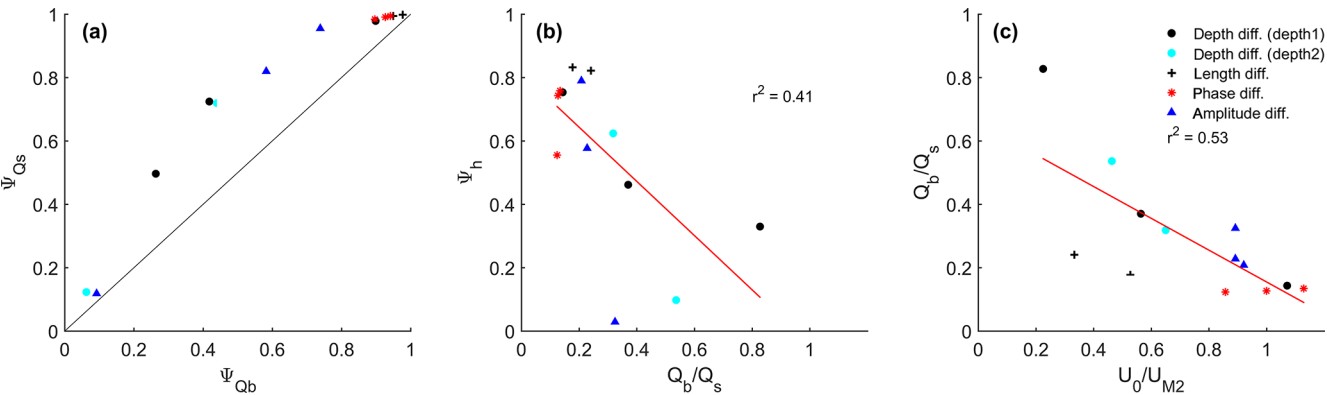

**Figure 11.** Comparisons of **(a)** suspended load asymmetry ($\Psi_{\text{susp load}}$) against bedload asymmetry ($\Psi_{\text{bedload}}$) overlaid with the line of equality (black line), **(b)** scatter plot of morphology asymmetry ($\Psi_h$) against ratio of bedload and suspended load transport in the upstream channel, and **(c)** scatter plot ratio of bedload and suspended load transport in the upstream channel against the dominance of river flow over tidal flows in the upstream channels. The legend for all panels is provided in panel **(c)**.

load. As a result, the depth asymmetry between downstream channels increases. Similarly, a coarser sediment results in smaller depth asymmetry between the downstream channels.

The importance of the effect of lateral bed slope to oppose the asymmetrical morphological development between downstream channels causes the model results to be sensitive to the parameter $\alpha_{\text{bn}}$. Using physical scale models, previous studies have suggested that $\alpha_{\text{bn}}$ should take values between 0.2 and 1.5 (Baar et al., 2018; Ikeda, 1982; Schuurman et al., 2013; Talmon et al., 1995). However, Delft3D shows unrealistic morphological development when small values of $\alpha_{\text{bn}}$ are used, as shown in Fig. 13. Simulation Depth1_Q2800 with small $\alpha_{\text{bn}}$ ($\alpha_{\text{bn}} = 1$) showed the development of an elongated bar upstream on the side of the shallow downstream channel and a large incision occurred on the other side. This unrealistic behaviour has also been evaluated by Baar et al. (2019). The use of a small value for $\alpha_{\text{bn}}$ causes the morphological development to be dependent on the grid size (Baar et al., 2019). Several studies have used much higher values to overcome this issue (e.g. Dissanayake et al., 2009; van der Wegen and Roelvink, 2008, 2012). Using our model set-up, the model results started to be insensitive to the value of $\alpha_{\text{bn}}$ when $\alpha_{\text{bn}}$ is 10. Using this value, the lateral slope developing upstream of the bifurcation is less than 3 times the upstream channel width as also suggested by Bolla Pittaluga et al. (2003) and Kleinhans et al. (2008) for river-dominated bifurcations.

### 4.4 Implications of results

From the findings presented in this paper, we can predict how tides will influence the morphological evolution of deltas. In the seaward part of tide-influenced deltas, especially those with seaward-widening channels, river flow tends to be small relative to the tidal flows. In these regions we only expect asymmetry in morphology when the branches are unequally

forced by tides. The tides tend to keep all the branches open and have similar depths. In the upstream part of deltas, river flows tend to be larger, which can result in large morphological asymmetries. However, the different possible pathways of the tide along the channel networks can generate differences in tidal amplitude and tidal phase between branches, inducing relatively strong tidal currents at the junction. This prevents the closure of one downstream channel and erodes the bed at the junction because of the strong cross-channel flows.

Morphological development of bifurcations occurs on a long timescale and several external causes and internal processes neglected here can affect bifurcation stability (also see the review in Kleinhans et al., 2013), such as of sea level rise (Jerolmack, 2009; Van Der Wegen, 2013), changes in upstream discharge or sediment supply (Syvitski and Milliman, 2007), channel bank erosion or growth (Miori et al., 2006), and delta front development that could change the length of a branch (Salter et al., 2017). However, we have provided a basic explanation on how tides can stabilize the morphology of deltas.

## 5 Conclusions

In this article, the effect of tides on the morphological development of bifurcations was investigated using a numerical modelling approach in Delft3D. An idealized bifurcation was built by splitting an upstream channel into two downstream branches. The idealized bifurcations were forced by river discharge from upstream and tides from downstream. To identify the effect of tides, two cases were studied, namely geometry difference (length and depth of channels) and tide difference (difference in prescribed tides at the two downstream channels).

The results show that an increased tidal influence compared to river influence results in a less asymmetric mor-

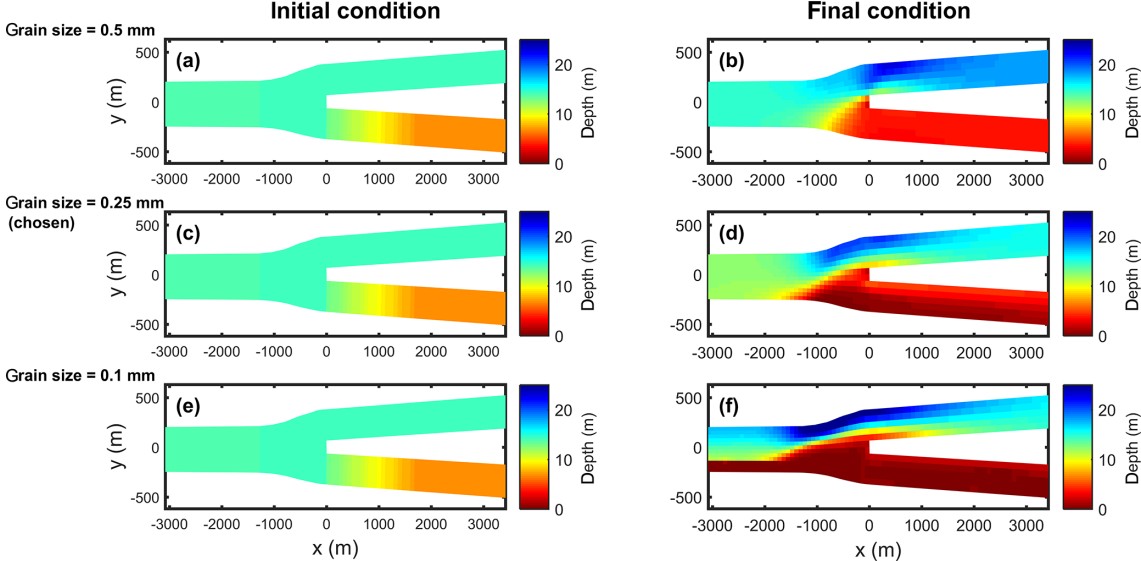

**Figure 12.** Initial **(a, c, e)** and final **(b, d, f)** depth near the bifurcation for coarser sand **(a, b)**, applied sand **(c, d)**, and finer sand **(e, f)** using the set-up of simulation Depth1_Q2800.

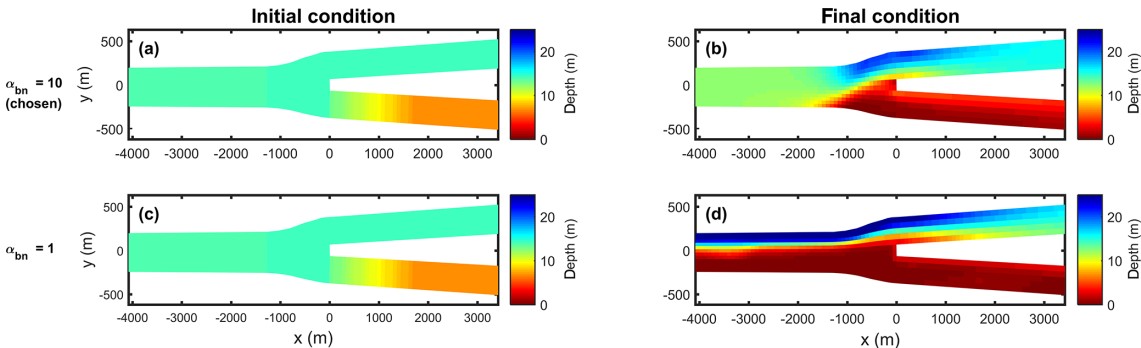

**Figure 13.** Initial **(a, c)** and final **(b, d)** depth near the bifurcation for large **(a, b)** and small **(c, d)** $\alpha_{bn}$ using the set-up of simulation Depth1_Q2800.

phology of the bifurcation. This increased tidal influence can be achieved either by smaller river discharge or by asymmetric tides from downstream. The main mechanism is that tidal flows tend to be less asymmetric in the two downstream channels than tidally averaged flows. This causes the peak Shields number in the branches to be closer to each other with increasing influence of tides. Furthermore, we have shown that bedload transport tends to be divided less asymmetrically than suspended load due to the influence of lateral bed slopes, which tends to stabilize the system. In our simulations, bifurcations with increased tidal influence had a relatively high ratio of bedload over suspended load transport and therefore developed a less asymmetric morphology than in river-dominated systems. Our results can explain why tides tend to stabilize the bifurcations in deltas.

## Appendix A

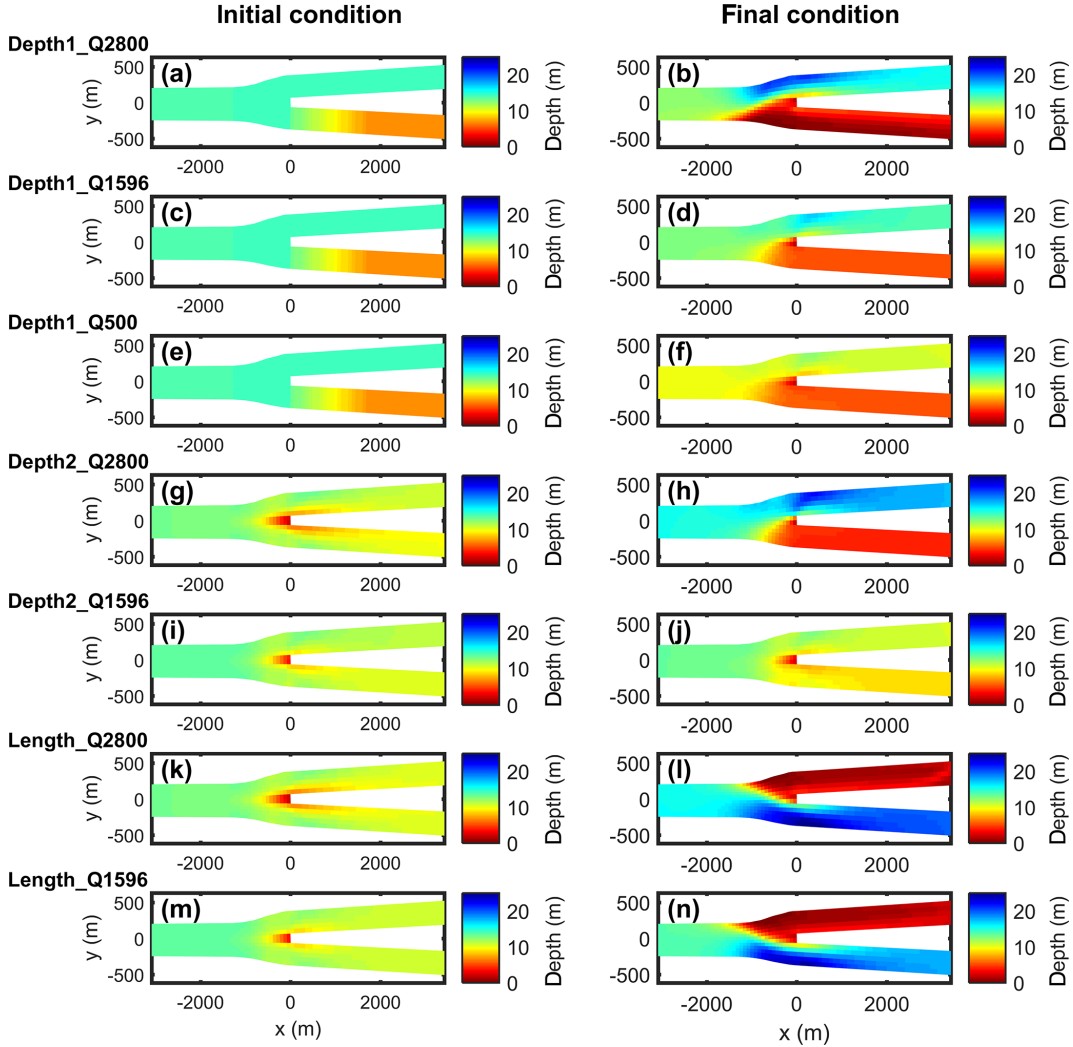

**Figure A1.** Initial (left panels) and final (right panels) depth near the bifurcation for all simulations in Case 1. For the *depth difference* scenario **(a–j)**, the top branch in each panel is the deep downstream branch and the bottom one is the shallow downstream branch. For the *length difference* scenario **(k–n)**, the top branch in each panel is the long downstream branch and the bottom branch is the short downstream branch.

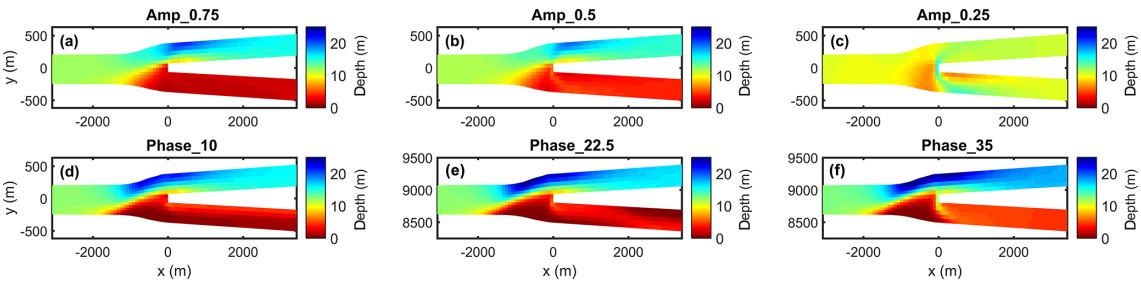

**Figure A2.** TS6 Final depth for Case 2. For the *amplitude difference* scenario **(a–c)**, the downstream branch imposed by low tides is the bottom branch, while for the *phase difference* scenario **(d–f)** the bottom branch is the downstream branch imposed by delayed tides.

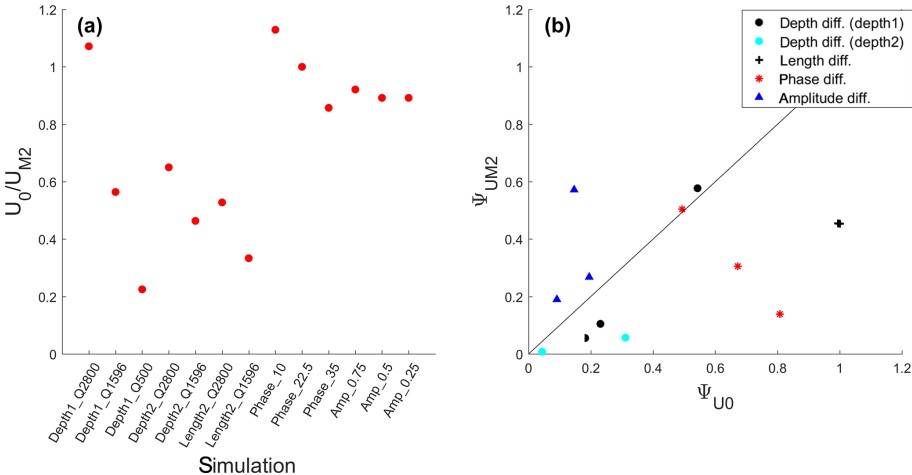

**Figure A3. (a)** Ratio of $U_0$ and $U_{M_2}$ for all simulations and **(b)** comparison of asymmetry of $U_{M_2}$ against asymmetry of $U_0$.

**Table A1.** Sediment mobility (tide-averaged and maximum), mean flow, and tidal flow amplitude at the cross sections near the bifurcation as shown in Fig. 2 for all simulations. Main channel is the upstream channel, minor branch is the downstream channel that tends to be shallower, and major branch is the deepened downstream channel.

| Simulation | Mobility ($\tau_{*ave}$) | | | Mobility ($\tau_{*max}$) | | | $U_0$ | | | $U_{M_2}$ | | |
|---|---|---|---|---|---|---|---|---|---|---|---|---|
| | Main channel | Minor branch | Major branch | Main channel | Minor branch | Major branch | Main channel | Minor branch | Major branch | Main channel | Minor branch | Major branch |
| Depth1_Q2800 | 0.76 | 0.04 | 0.28 | 0.66 | 0.14 | 0.67 | 0.30 | 0.16 | 0.54 | 0.28 | 0.14 | 0.52 |
| Depth1_Q1596 | 0.28 | 0.09 | 0.16 | 0.29 | 0.24 | 0.44 | 0.22 | 0.20 | 0.32 | 0.39 | 0.42 | 0.52 |
| Depth1_Q500 | 0.07 | 0.07 | 0.09 | 0.20 | 0.18 | 0.26 | 0.09 | 0.09 | 0.13 | 0.40 | 0.45 | 0.50 |
| Depth2_Q2800 | 0.13 | 0.10 | 0.19 | 0.37 | 0.29 | 0.50 | 0.26 | 0.21 | 0.40 | 0.40 | 0.45 | 0.50 |
| Depth2_Q1596 | 0.10 | 0.11 | 0.12 | 0.27 | 0.30 | 0.31 | 0.19 | 0.22 | 0.24 | 0.41 | 0.49 | 0.50 |
| Length2_Q2800 | 0.11 | 0.02 | 0.22 | 0.31 | 0.08 | 0.60 | 0.19 | 0.00 | 0.44 | 0.31 | 0.20 | 0.53 |
| Length2_Q1596 | 0.08 | 0.02 | 0.15 | 0.24 | 0.08 | 0.45 | 0.12 | −0.01 | 0.29 | 0.32 | 0.21 | 0.55 |
| Phase_10 | 0.15 | 0.04 | 0.28 | 0.40 | 0.16 | 0.67 | 0.30 | 0.18 | 0.53 | 0.31 | 0.17 | 0.55 |
| Phase_22.5 | 0.15 | 0.04 | 0.28 | 0.38 | 0.16 | 0.66 | 0.26 | 0.10 | 0.51 | 0.36 | 0.30 | 0.56 |
| Phase_35 | 0.17 | 0.07 | 0.27 | 0.41 | 0.20 | 0.68 | 0.22 | 0.05 | 0.47 | 0.45 | 0.47 | 0.61 |
| Amp_0.75 | 0.17 | 0.10 | 0.26 | 0.42 | 0.23 | 0.66 | 0.35 | 0.35 | 0.47 | 0.31 | 0.16 | 0.58 |
| Amp_0.5 | 0.19 | 0.11 | 0.26 | 0.49 | 0.37 | 0.66 | 0.33 | 0.31 | 0.46 | 0.40 | 0.35 | 0.61 |
| Amp_0.25 | 0.23 | 0.20 | 0.25 | 0.54 | 0.58 | 0.59 | 0.33 | 0.42 | 0.35 | 0.51 | 0.49 | 0.71 |

**Code availability.** The model set-up for all Delft3D simulations is provided in the Supplement. The results presented were simulated using the Delft3D software package (Delft3D-flow version 4.01.00.rc.04).

**Supplement.** The supplement related to this article is available online at: https://doi.org/10.5194/esurf-8-1-2020-supplement.

**Author contributions.** API, MvdV, and MGK designed the study. API conducted the numerical modelling, performed the output analysis, and interpretation and wrote the article with major input from MvdV and MGK. MvdV conducted part of the output analysis and edited the paper.

**Competing interests.** The authors declare that they have no conflict of interest.

**Acknowledgements.** This research is funded by an Indonesia Endowment Fund for Education (LPDP) grant to Arya P. Iwantoro.

**Financial support.** This research has been supported by the Indonesia Endowment Fund for Education (LPDP) (grant no. 20161222029838). TS7

**Review statement.** This paper was edited by Patricia Wiberg and reviewed by John Shaw and one anonymous referee.

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

**Remarks from the language copy-editor**

**Remarks from the typesetter**

**TS2** The equation was approved like this and should not be adjusted without the editor's approval. We have also not adjusted the requested values. Meaning and content changes, including changes to values, should be reviewed by the editor before being implemented in the proofreading stage. Please reassess if these changes are strictly necessary before taking this step. For more information, please see our proofreading guidelines on the AMT website: https://www.earth-surface-dynamics.net/for_authors/proofreading_guidelines.html.