# Peer review of "Morphological evolution of bifurcations in tide-influenced deltas"

_Earth Surface Dynamics, 2019_

## Referee Comment (RC1) · Anonymous Referee #1 · 30 Dec 2019

The manuscript presents a study on tide-influenced bifurcations based on morphodynamic modelling. The modelling is carried out for a schematized bifurcation and aims to understand tides influence the morphological development of bifurcations. This is an important subject and the conclusions are relevant for understanding the morphological development of tide-influenced deltas. I support publication of the manuscript after moderate revision.

I think that the manuscript can be improved by presenting more thorough analysis of the model results. The additional analysis should take away the vagueness in the conclusions, like those formulated in the abstract:

- Line 18-19. "…our results show that bedload tends to divide less asymmetrical compared to suspended load, showing a **_possible_** stabilizing effect of lateral bed slopes on morphological evolution." The word "possible" suggest that the authors are not sure about this. Better analysis of the model results should clarify this.
- Line 19-20. "In our simulations, the more tide-dominated systems tend to have a larger ratio of bedload and suspended load transport." How should I read this? Is this a general conclusion, or is it just because of some special feature in your simulations? In the last case it is not worthy to mention in the abstract, unless it gives explanation to the other conclusions. Otherwise you need to give the physical mechanisms explaining it.

In the model set-up some of the parameters have been given a fixed value without sufficient motivation: "horizontal eddy viscosity was set to 10 $m^2s^{-1}$", "value of 10 for $\alpha_{bn}$", "$\alpha_{bs} =1$" (Line 102-106). Especially $\alpha_{bn}$ is a key parameter influencing the distribution of sediment transport to the two downstream branches. Also the horizontal eddy viscosity may be important for the local flow pattern around the bifurcation. Therefore, I expected some sensitivity analysis on these parameters, or at least some motivation why fixed values for them can be used in the study without influencing the conclusions.

Line 134. "to have Courant Number smaller than 1", why is this needed? I thought that Delft3D uses an implicit scheme.

Line 143 & Section 2.2. How about the morphological boundary conditions? What was prescribed at the e.g. the upstream boundary, sediment transport rate or fixed bed?

Line 153. Note that even for the largest discharge (2800 $m^3$/s) the velocity at the upstream boundary is only about 0.5 m/s.

Line 178. I do not understand immediately why the first 2 km determines the morphological development of the entire downstream channel.

Line 211. "but the depth of the two downstream channels does not depend on the discharge", this is remarkable. I wonder if this is not because of the short simulation time. Influence of the upstream boundary not yet reached to the downstream branches?

Details:

Line 183. Change "duration of simulations" to "simulated period"?

Line 192. "$m^3$" should be "$m^{-3}$".

Line 195. Eq. (4) not needed, state the similar (to Tau) definitions for other parameters.

Line 238. "Length" should be "length".

---

## Referee Comment (RC2) · John Shaw (Referee) · 6 Jan 2020

John Shaw (Referee)

shaw84@uark.edu

This manuscript describes a modeling study that seeks to understand morphodynamic adjustments to bifurcations that occur due to river and tide interaction. A novel set of boundary conditions is used. The key finding is that as tidal forcing or tidal heterogeneity (the use phase lags) increases, the stability or symmetry of the bifurcation increases through adjustments to the sediment bed. The straightforward modeling approach affords a relatively clear view of the controlling processes. It is well written, and a solid improvement to our understanding of river delta networks.

One important question I have after digesting this manuscript is the following (the subject heading): The explanation of flow regulation is that the tidal flow from the bigger

[Figure]

channel pushes flow into the smaller channel (L289, reason one). This sounds like a rising tide phenomenon, where an incoming tidal waves hits the bifurcation from downstream. However, this is probably a relatively low shear stress and sediment transport moment as the tide fights with the river. The paper states that the most symmetric shields stresses occur at peak ebb flow (L264). At this time, I would expect a falling tide in a deeper channel would pull more water and be more asymmetric. Perhaps this could be elucidated with a Psi_tau*max plot over a tidal period? It would help me understand this key aspect of the system.

Figure 8-11 show correlations of various strength between asymmetry and modeling runs. I am quite surprised by the scatter for example in Fig. 9. With a model that is so simply designed, I would like to know where the scatter comes from. Even if the authors suspect is from numerical errors, it would be good to know. The authors' intuition for this is far better than mine (or the average reader). No information was given about initial bed elevation or bed slope, which seems like an important boundary condition for tidal waves and backwater dynamics. If the system was initialized with a uniform elevation, that information will suffice. Modeling 0.25 mm sediment (L104) in a large tidal system seems too large to be characteristic of tidal systems. I am not suggesting to redo the modeling, but can you justify this choice further?

Minor Comments L110 I appreciate the discussion of the limitation of the non-adjustible widths. I think it is a reasonable simplifying assumption for this study though. L124 within 800 m of L152-153 over 2km is redundant L166 It took some effort to figure out what eta_i means. I found it in figure 1, but perhaps it could also be explicitly defined in the text here. L234 The Chezy friction factor was set to be constant, so I do not see how differing friction could matter here. Varying depth and the associated reduction in tidal wave celerity is a much more intuitive explanation here. L289 Processes instead of reasons? L326 Findings L351 asymmetrically L354 "relatively ratio" a word is missing here. Figure 2-7. The size and colormaps make these figures very difficult to gather information from. I recommend either adding 2-4 contours to the plots or just using 2-4

colors instead of a spectrum. Basically all of the detail in plots like Figures like Fig7a can't be seen by the reader.

---

## Short Comment (SC1) · 9 Jan 2020

Dear Editor,

here I provide comments on several specific points:

1) I think the authors should take more in consideration some of the recent theoretical results.

The analytical model of Redolfi et al. (2016), which represents a fully two-dimensional extension of the Bolla Pittaluga et al. (2003, 2015) model, has been proven to effectively predict the stability of both gravel and sand bed bifurcation (Redolfi et al. 2019) with the key advantage of avoiding the calibration of a specific parameter (like the $\alpha$ parameter of Bolla Pittaluga el al., 2003). I think these recent advancements should be

at least mentioned in the Introduction and/or in the Discussion (Section 4.2).

Similarly, in the the work of Salter et al. (2017) which specifically focuses on the effect of downstream conditions on the bifurcation stability, should be considered in the Introduction.

Please ignore this comment if you think it is too personal (I am the main author of the above-mentioned papers).

2) Line 52: this sentence is misleading, as Bolla Pittaluga et al. (2003) did not test the effect of meandering bends. 3) Line 97: the sentence "the 2D approach also results in reliable morphodynamic simulations" is very vague, as the reliability of depth-averaged models clearly depends on the specific problem under consideration.

4) Section 4.2: it would be useful to provide an indication of the average Shields number.

5) Code availability: to enable the reproducibility of the results, I recommend the authors to share the configuration files.

6) System of coordinates: I can not see the reason for placing the bifurcation at x=220 km an y~9 km. What is the meaning of the origin (i.e. x=0, y=0) point? Setting x=0 and y=0 at the bifurcation node would have been more meaningful, and it would have facilitated the reading of the maps.

7) Units: space between value and unit is sometimes missing (e.g., line 177); please also note that units should not appear in italic (Equation 1);

8) Graphics: in several figures (e.g., Figures 7 and 11) the labels are disproportionately small; in Figure A1: the quantities represented on the two axes are "x" and "y", not "x-axis" and "y-axis.

Sincerely,

Dr. Marco Redolfi

**ESurfD**

---

## Author Comment (AC3) · 6 Mar 2020

We thank the Referee for his time to carefully read the manuscript and for his comments that helped us improving the manuscript.

1. **Comment from referee:**

   1) I think the authors should take more in consideration some of the recent theoretical

   results.

   The analytical model of Redolfi et al. (2016), which represents a fully two-dimensional extension of the Bolla Pittaluga et al. (2003, 2015) model, has been proven to effectively predict the stability of both gravel and sand bed bifurcation (Redolfi et al. 2019) with the key advantage of avoiding the calibration of a specific parameter (like the α parameter of Bolla Pittaluga el al., 2003). I think these recent advancements should be at least mentioned in the Introduction and/or in the Discussion (Section 4.2).

   Similarly, in the the work of Salter et al. (2017) which specifically focuses on the effect of downstream conditions on the bifurcation stability, should be considered in the Introduction.

   Please ignore this comment if you think it is too personal (I am the main author of the above-mentioned papers).

   **Author's response:**

   We agree that the mentioned work is a good extension of the work of Bolla Pittaluga et al (2003; 2015), although the fundamentals are still the same. Furthermore, we are aware of the work of Salter et al. (2017).

   **Author's changes in manuscript:**

   We will add these references in both introduction and discussion. We did not fully compare out model results with the mentioned papers because by using a 2DH model we don't need a nodal point relation.

2. **Comment from referee:**

   2) Line 52: this sentence is misleading, as Bolla Pittaluga et al. (2003) did not test the effect of meandering bends. 3) Line 97: the sentence "the 2D approach also results in reliable morphodynamic simulations" is very vague, as the reliability of depth-averaged models clearly depends on the specific problem under consideration.

   **Author's response:**

   Line 52: We agree that the text is not very clear in the previous version. We meant that the transverse channel slope, as studied by Kleinhans et al. (2008), was similar as in Bolla Pittaluga et al. (2003). However, Kleinhans et al. (2008) added the effect of meandering upstream channel in the transverse transport calculation.

   Line 97: Since in this study we are interested to investigate a large scale morphological development, simulating a detail 3D result is not our goal. Therefore, 2D model is sufficient and can reduce the computational cost.

   **Author's changes in manuscript:**

Line 52: We will remove the citation to Bolla Pittaluga et al. (2003) for this sentence to avoid misunderstanding.

Line 97: In Section 2.1 (Model set-up) in the new version, we will strengthen and elaborate our argument why we chose 2D model instead of 3D according to the reasoning that we mention in the Author response.

3. **Comment from referee:**

4) Section 4.2: it would be useful to provide an indication of the average Shields number.

**Author's response:**

We agree. We will provide a table of Shields numbers and flow conditions at the bifurcation.

**Author's changes in manuscript:**

We will make a table that provides the information of: tide averaged Shields number, maximum Shields number, mean flow velocity, and $M_2$ velocity amplitude at the bifurcation in Appendix.

4. **Comment from referee:**

5) Code availability: to enable the reproducibility of the results, I recommend the authors to share the configuration files.

**Author's response:**

We will provide the configuration files for all simulations in the supplement.

**Author's changes in manuscript:**

The supplement will be mentioned in the new version.

5. **Comment from referee:**

6) System of coordinates: I cannot see the reason for placing the bifurcation at x=220 km an y~9 km. What is the meaning of the origin (i.e. x=0, y=0) point? Setting x=0 and y=0 at the bifurcation node would have been more meaningful, and it would have facilitated the reading of the maps.

**Author's response:**

This coordinate system was due to the way the grid was built in which x=0 is located at the upstream end of upstream channel.

**Author's changes in manuscript:**

We will move the origin of the x and y coordinate system to the bifurcation in the new version.

6. **Comment from referee:**

7) Units: space between value and unit is sometimes missing (e.g., line 177); please also note that units should not appear in italic (Equation 1);

**Author's response and changes in manuscript:**

The space will be added in the new version for all missing space. Equation 1 will be also fixed.

**7. Comment from referee:**

8) Graphics: in several figures (e.g., Figures 7 and 11) the labels are disproportionately small; in Figure A1: the quantities represented on the two axes are "x" and "y", not "x-axis" and "y-axis.

**Author's response and changes in manuscript:**

The labels and the axes will be improved according to this comment in the new version.

---

## Author Response (AR1)

We thank the associate editor for helping us to improve the manuscript. In this document we provide the author's response to the referees and the marked up version of the manuscript.

**Referee 1**

The manuscript presents a study on tide-influenced bifurcations based on morphodynamic modelling. The modelling is carried out for a schematized bifurcation and aims to understand tides influence the morphological development of bifurcations. This is an important subject and the conclusions are relevant for understanding the morphological development of tide-influenced deltas. I support publication of the manuscript after moderate revision.

We thank the Referee for his time to read the paper. It helped us to improve the manuscript.

1. **Comment from referee:**

   I think that the manuscript can be improved by presenting more thorough analysis of the model results. The additional analysis should take away the vagueness in the conclusions, like those formulated in the abstract:

   • Line 18-19.

   "…our results show that bedload tends to divide less asymmetrical compared to suspended load, showing a possible stabilizing effect of lateral bed slopes on morphological evolution." The word "possible" suggest that the authors are not sure about this. Better analysis of the model results should clarify this.

   **Author's response:**

   We agree that "possible" is too weakly formulated. We already know that a lateral bed slope has a stabilizing effect for a bifurcation (Bolla Pittaluga et al., 2003). They showed that the lateral bed slope only affects the bedload transport, causing additional transport of sediment towards the deeper channel. Such a compensating mechanism is not present for suspended load transport. Bedload tend to divide less asymmetric than suspended load as we have shown in the manuscript in Figure 11a below, also because suspended load has a stronger dependence on flow velocity than bed load transport.

[Figure]

Figure 11a from the manuscript. Comparison of suspended load asymmetry ($\Psi_{susp\ load}$) against bedload asymmetry ($\Psi_{bedload}$) overlaid by the line of equality (black line),

**Author's changes in manuscript:**

In the abstract of the revised version we removed the word "possible" to be more clear. The previous statement:

"…our results show that bedload tends to divide less asymmetrical compared to suspended load, showing a possible stabilizing effect of lateral bed slopes on morphological evolution."

Our improvement is in Line 18-20 (Line 18-20 in the marked up version below) as follows:

"…our results show that bedload tends to divide less asymmetrical compared to suspended load and confirm the stabilizing effect of lateral bed slopes on morphological evolution as was also found in previous studies."

2. **Comment from referee:**

• Line 19-20. "In our simulations, the more tide-dominated systems tend to have a larger ratio of bedload and suspended load transport." How should I read this? Is this a general conclusion, or is it just because of some special feature in your simulations? In the last case it is not worthy to mention in the abstract, unless it gives explanation to the other conclusions. Otherwise you need to give the physical mechanisms explaining it.

**Author's response:**

According to our simulations with different asymmetry in geometry and tides, we have a consistent behaviour where a larger ratio of bedload and suspended load transport occurs in the more tide-dominated condition as given in Fig. 11c below. This is because the tide-river interaction causes a long period of weak flows in one tidal cycle, the strongest flow occurs during ebb tides at which the suspended load transport is the highest. Since river discharge dampens the tides, the low sediment mobility condition which favour bedload transport occurs for a relatively long period, which is during flood and during transition between ebb and flood tides.

[Figure]

Figure 11 from the manuscript. Comparison of: (a) Suspended load asymmetry ($\Psi_{\text{susp load}}$) against bedload asymmetry ($\Psi_{\text{bedload}}$) overlaid by the line of equality (black line), (b) Scatter plot of morphology asymmetry ($\Psi_h$) against ratio of bedload and suspended load transport in the upstream channel, and (c) Scatter plot ratio of bedload and suspended load transport in the upstream channel against the dominance of river flow over tidal flows in the upstream channels. The legend for all panels is provided in panel c.

**Author's changes in manuscript:**

We added additional explanation in Line 20-21 in the revised version (Line 20-23 in the marked up version below) to clarify this as follows:

"We show that the more tide-dominated systems tend to have a larger ratio of bedload and suspended load transport due to periodical low sediment mobility conditions during a transition between ebb and flood. "

**3. Comment from referee:**

In the model set-up some of the parameters have been given a fixed value without sufficient motivation: "horizontal eddy viscosity was set to 10 m2s-1", "value of 10 for αbn", "αbs =1" (Line 102-106). Especially αbn is a key parameter influencing the distribution of sediment transport to the two downstream branches. Also the horizontal eddy viscosity may be important for the local flow pattern around the bifurcation. Therefore, I expected some sensitivity analysis on these parameters, or at least some motivation why fixed values for them can be used in the study without influencing the conclusions.

**Author's response:**

For horizontal eddy viscosity, the chosen value (10 $m^2s^{-1}$) was chosen because a small value can cause numerical instability in the model near the bifurcation because flow magnitudes and direction quickly change. It does not influence the final results.

Transverse bed slope effects for bedload transport were accounted for by the approach of Ikeda (1982) and we used a value of 10 for $\alpha_{bn}$. This value is much higher than the default value (1.5) and values suggested by Bolla Pittaluga et al. (2003) (0.3-1) because a small value of this parameter in Delft3D leads to unrealistic and grid size-dependent channel incision (Baar et al., 2019). This is kind of a model artefact. The model 'needs' a diffusive process. Near the mouth of an estuary this can come from waves (Ridderinkhof et al. (2016) used realistic values of $\alpha_{bn}$). Note that although the value is large, it is well within the range of what others used (e.g. Dissanayake et al., 2009; van der Wegen and Roelvink, 2012; Van Der Wegen and Roelvink, 2008). However, we agree that it is an important parameter and that sensitivity should be studied.

For streamwise bed slope effects the Bagnold (1966) approach was used with a Delft3D default value of $\alpha_{bs}$ =1. In general, this does not have a large impact on modelled bed evolution.

**Author's changes in manuscript:**

The reasoning for the chosen horizontal eddy viscosity was added in Line 112-114 in the revised version (Line 114-116 in the marked up version). For the transverse bedslope effect, additional reasoning mentioned in the Author's response was added in Line 119-123 (Line 121-125 in the marked up version). To strengthen the argument, we ran an additional simulation that apply the $\alpha_{bn}$ of 1 and compared it with the one we used for all simulations ($\alpha_{bn}$ = 10). Compared simulations had the same settings except for $\alpha_{bn}$ to clearly see the effect of the chosen parameter. The results are shown in the plot of final morphology for the two simulations below. The difference between the results from these simulations was discussed in Line 369-380 (Line 376-387 in the marked up version below) and the figure was shown in Line 583 (Line 601 in the marked up version). For streamwise bedslope, we will mention that we used the default value from Delft3D in Line 123-124 (Line 125-126 in the marked up version).

[Figure]

Figure: Initial (left panels) and final (right panels) depth near the bifurcation for different $\alpha_{bn}$. The same initial and boundary conditions were prescribed as the simulations for the sensitivity of grain size.

4. **Comment from referee:**

Line 134. "to have Courant Number smaller than 1", why is this needed? I thought that Delft3D uses an implicit scheme.

**Author's response:**

According to Lesser et al. (2004), Delft3D applies an alternating direction implicit (ADI) to solve the continuity and momentum balance in hydrodynamic simulations. In this scheme one direction is solved implicitly, the other one explicitly. This alternates between time steps. This approach requires a certain courant number for the numerical stability reason (Lesser et al., 2004). Though it has been stated in several papers (e.g. Long et al., 2008; Reyns et al., 2014) that best results are obtained for Courant <1, especially for morphodynamic simulations and system with steep bends, in the Delft3D manual (Version: 3.15.29178, 17 July 2013), it is stated that using ADI scheme the maximum Courant number to fulfil the numerical stability is $4\sqrt{2}$ while our model has the maximum Courant number below 1 which fulfil this requirement.

**Author's changes in manuscript:**

In Line 110-111 in the revised version (Line 112-113 in the marked up version), we mentioned the numerical scheme to solve the shallow water equations. We mentioned the maximum Courant number to fulfil the numerical stability is 4√2 in Line 150-151 (Line 152-153 in the marked up version).

5. **Comment from referee:**

Line 143 & Section 2.2. How about the morphological boundary conditions? What was prescribed at the e.g. the upstream boundary, sediment transport rate or fixed bed?

**Author's response:**

At inflow condition, the equilibrium sediment transport was prescribed. Thus, the morphology at the open boundaries does not change. During outflow, no boundary condition was needed at the open boundaries and the bed was free to evolve.

**Author's changes in manuscript:**

We explained morphological boundary condition in Line 154-156 in the revised version (Line 157-159 in the marked up version) as mentioned in the Author's response.

6. **Comment from referee:**

Line 153. Note that even for the largest discharge (2800 m3/s) the velocity at the upstream boundary is only about 0.5 m/s.

**Author's response:**

Because we built a typical setting for a tidal delta we prescribed a channel with gentle slope ($3 \times 10^{-5}$) (Line 161 in the revised version (Line 164 in the marked up version)). This drives a small river flow. Besides, we also need to prescribe this small river flow to let the tides propagate to the upstream channel and to have conditions in which tidal flows are larger than river flows.

**Author's changes in manuscript:**

-

7. **Comment from referee:**

Line 178. I do not understand immediately why the first 2 km determines the morphological development of the entire downstream channel.

**Author's response:**

From our simulations, we found that the development of the downstream channels starts from upstream and develops downstream. Therefore, analysing the most upstream end of the downstream channel is sufficient to determine the growth in asymmetry between them. Based on our simulations, a distance shorter than 2 km cannot be representative to determine the behaviour of the downstream channel (whether they are silting up or deepening) due to the presence of local morphological features near the bifurcation such as bar formation or small incisions in the downstream channel that is silting up. However, a longer distance cannot be representative to determine the downstream channel asymmetry or avulsion because even though one downstream channel almost avulses upstream, tides can cause a deepening near the downstream boundary as

shown in length difference scenario as shown in Figure 5c and f and phase difference scenario in figure 7e and h provided below. Hence, we decided to use the average bed level first 2 km as a representative depth.

[Figure]

Figure 5 from the manuscript. Time-stack diagram of width- and tide-averaged depth as a function of space for the simulations in Length difference scenario with the same order as Figure 4 but with short (panel (b) and (e)) and long (panel (c) and (f)) downstream branch.

[Figure]

Figure 7 from the manuscript. Time-stack diagram of width- and tide-averaged depth as a function of space for Phase difference scenario.

**Author's changes in manuscript:**

We explained our motivation to use the first 2 km in Line 197-204 in the new manuscript (Line 201-207 in the marked up version).

8. **Comment from referee:**

Line 211. "but the depth of the two downstream channels does not depend on the discharge", this is remarkable. I wonder if this is not because of the short simulation time. Influence of the upstream boundary not yet reached to the downstream branches?

**Author's response:**

The statement in line 211 is describing the depth of the two control simulations namely Control_Q2800 and Control_Q1596 which is named according on the prescribed discharge upstream and are with 1 m tidal amplitude at the downstream boundaries. The morphology of the two downstream channels still depends on the prescribed discharge. However, the difference between those simulations is small as shown in time-stack figure below

[Figure]

Figure above is time-stack figure of the difference between the depth of simulation Control_Q2800 and Control_Q1596 in branch 1 (left) and branch 2 (right). Negative value means simulation Control_Q1596 is deeper than Control_Q2800. The depth from simulation Control_Q1596 at the end of the simulation is slightly deeper at the end of the simulation period. Here I also provide Figure 3 (the depth from the two compared simulations) from the manuscript.

[Figure]

Figure 3 from the manuscript. Time-stack diagram of width- and tide-averaged depth (colour) of the upstream channel (left panels; km 0 is junction, km 20 upstream) and downstream channels (middle and right panels; km 0 is junction, km 30 near sea) as a function of distance from the bifurcation (vertical axis) for the two control simulations. The top panels ((a), (b), and (c)) are the result from the high discharge simulation (Control_Q2800) while the bottom panels ((d), (e), and (f)) are for the low discharge simulation (Control_Q1596).

This small difference in the downstream channel between the two control simulations is because a strong influence of tides in the downstream channels in controlling the morphology. Though river discharge dampens the tides, the widening channel aids to maintain the tidal flow upstream in both downstream channels and causes river-induced flow velocities to become small. Since the two simulations are imposed by the same tidal amplitude, a similar morphology occurs in the downstream channels for both simulations

**Author's changes in manuscript:**

In the previous version we stated: "but the depth of the two downstream channels does not depend on the discharge".

In Line 238-240 in the revised version (Line 241-243 in the marked up version) we clarified it:

"...but the depth of the two downstream channels does not significantly affect the depth of the two downstream channels. This is because both control simulations were imposed by the same tidal forcing and the morphology of the downstream channels is mainly controlled by the tides."

Details:

9. **Comment from referee:**

Line 183. Change "duration of simulations" to "simulated period"?

Line 192. "m3" should be "m-3".

**Author's response and changes in manuscript:**

We improved them in the revised version based on these comments (Line 208 and Line 217 in the revised version (Line 211-220 in the marked up version), respectively).

**Referee 2**

This manuscript describes a modeling study that seeks to understand morphodynamic adjustments to bifurcations that occur due to river and tide interaction. A novel set of boundary conditions is used. The key finding is that as tidal forcing or tidal heterogeneity (the use phase lags) increases, the stability or symmetry of the bifurcation increases through adjustments to the sediment bed. The straightforward modeling approach affords a relatively clear view of the controlling processes. It is well written, and a solid improvement to our understanding of river delta networks.

We thank the Referee for his time to carefully read the manuscript and for his comments that helped us improving the manuscript.

1. **Comment from referee:**

   One important question I have after digesting this manuscript is the following (the subject heading): The explanation of flow regulation is that the tidal flow from the bigger channel pushes flow into the smaller channel (L289, reason one). This sounds like a rising tide phenomenon, where an incoming tidal waves hits the bifurcation from downstream. However, this is probably a relatively low shear stress and sediment transport moment as the tide fights with the river. The paper states that the most symmetric shields stresses occur at peak ebb flow (L264). At this time, I would expect a falling tide in a deeper channel would pull more water and be more asymmetric. Perhaps this could be elucidated with a Psi_tau*max plot over a tidal period? It would help me understand this key aspect of the system.

   **Author's response:**

   To clarify, in line 264 in the previous version, $\Psi_{\tau*max}$ is the difference/asymmetry between the maximum peak Shields stress in the two downstream channels (which does not necessarily occur at the same time). This parameter has the strongest correlation with the asymmetry of the final depth between downstream channels.

   We also plotted the $\Psi_{\tau*}$ as a function of time for one of the simulations (simulation Depth1_Q500). If we compare the asymmetry during ebb and flood tides, we agree that the asymmetry during ebb tides is larger than during flood tides. However, the highest asymmetry occurs during the transition between ebb and flood as shown in Figure A below. Compared to this transition condition, the asymmetry during ebb tides is much smaller, even though flow velocities and the absolute difference between the two channels are largest. So, the relative difference decreases although the absolute difference increases. This also explains why for largest tidal influence the relative difference of the Shields stress is smallest. Furthermore, during flood tides the sediment mobility in the downstream channels is relatively small and sometimes close to the critical threshold of sediment motion. It means that during flood almost no sediment is transported. For the simulations with higher discharge (2800 and 1596 $m^3s^{-1}$) the sediment mobility during flood tides is below the critical threshold of sediment motion indicating no sediment transport.

[Figure]

Figure A: Flow velocity in the upstream channel two grids away from the junction (160 m) (blue line), and Shields stress in deep channel (red line) and shallow channel (yellow line) located near the bifurcation (160 m or two grids away) for simulation Depth1_Q500 (simulation imposed by depth difference between branches) overlaid by $\Psi_{\tau*}$ in time. The positive flow indicates ebb flow while negative is flood flow.

**Author's changes in manuscript:**

-

**2. Comment from referee:**

Figure 8-11 show correlations of various strength between asymmetry and modeling runs. I am quite surprised by the scatter for example in Fig. 9. With a model that is so simply designed, I would like to know where the scatter comes from. Even if the authors suspect is from numerical errors, it would be good to know. The authors' intuition for this is far better than mine (or the average reader). No information was given about initial bed elevation or bed slope, which seems like an important boundary condition for tidal waves and backwater dynamics. If the system was initialized with a uniform elevation, that information will suffice. Modeling 0.25 mm sediment (L104) in a large tidal system seems too large to be characteristic of tidal systems. I am not suggesting to redo the modeling, but can you justify this choice further?

**Author's response:**

The scatter in Fig 9 is because we use 5 asymmetry conditions: Depth1, Depth2, Length difference, tidal amplitude and phase difference. Most simulations started out of equilibrium because we do not know the morphodynamic equilibrium for these conditions. We picked these different scenarios to analyse how these types of imposed asymmetry, which is typically found in tidal deltas, will affect the morphodynamic evolution of the bifurcations. Even though we successfully found that they have a similar behaviour, i.e. more tidal influence drives a less morphological asymmetry between downstream channels, the quantity of the morphological asymmetry is different with different imposed asymmetry conditions. With changing morphology, the hydrodynamics also changed. So we did not have full control on where the system would finally end. Besides, not all simulations really reached morphological equilibrium. Though the

morphological change of all simulations at the end of the simulation is small, the morphological change in some simulations still develops very slowly.

Regarding, the channel slope, we have provided the streamwise channel slope ($3 \times 10^{-5}$) in previous version (line 142). Also, we provided the initial depth of all channels in Table 1.

Regarding the grain size, since we are interested in the effect of tides on the asymmetry of the bifurcation for different forcing conditions, we used a single value for the grain size. Thus, we chose a value representative for both more river- and more tide-dominated conditions. Medium sand is a good choice as it is also observed in some deltas such as in Berau River Delta (0.125-0.25 mm) (Buschman et al., 2013), Kapuas Delta (0.22-0.3 mm) (Kästner et al., 2017), Mahakam Delta (0.25-0.4 mm)  (Sassi et al., 2011), Mekong Delta (0.074-0.385 mm) (Stephens et al., 2017). However, we do agree that there is no single $D_{50}$ in a system and it differs between systems. The grain size influences the sediment transport mechanisms (bedload versus suspended load) and thereby effect the morphodynamics as shown in the figure below. Therefore, we will study the sensitivity of the model results to the value of $d_{50}$.

**Author's changes in manuscript:**

The scatter result in Figure 9 was further explained as mentioned in Author's response in in Line 318-322 in the revised version of the manuscript (Line 322-326 in the marked up version).

We motivated our choice of grain size as discussed above in Line 115-118 (Line 117-120 in the marked up version).

In Line 361-367 in the revised version (Line 368-374 in the marked up version), we discussed model results from a simulation with coarser grain size (0.5 mm) and from finer grain size (0.1 mm) using the same settings, asymmetry condition, and forcing as for the Depth1_Q2800 simulation (Table 1). These additional simulations showed how the model results were affected by the different type of grain size as shown below. Coarser sediment causes the morphology to be less asymmetric, because of the stabilizing effect of the lateral bed slope effect.

[Figure]

Figure B: Initial (left panels) and final (right panels) depth near the bifurcation for different prescribed grain size. Depth difference between downstream channels was prescribed. One downstream channel has a depth of 7.5 while the other channels were 15 m. At the upstream boundary the river discharge was 2800 $m^3s^{-1}$. At the downstream boundaries, $M_2$ tides were prescribed with the amplitude of 1 m.

**3. Comment from referee:**

Minor Comments L110 I appreciate the discussion of the limitation of the non-adjustible widths. I think it is a reasonable simplifying assumption for this study though. L124 within 800 m of L152-153 over 2km is redundant L166 It took some effort to figure out what eta_i means. I found it in figure 1, but perhaps it could also be explicitly defined in the text here.

**Author's response:**

The statement in Line 124 was about the width of the channels while the statement in line 152-153 is about the distance where the depth is gradually changing in the shallow downstream channel from the default depth (15 m) to 7.5 m for the first depth difference scenario (Depth1 in table 1). This depth change is gradual to avoid a sudden depth change that may affect the local flow condition.

**Author's changes in manuscript:**

We explained what eta_1 and eta_2 mean in Line 187-188 in the revised version (Line 190-191 in the marked up version) of the manuscript.

**4. Comment from referee:**

L234 The Chezy friction factor was set to be constant, so I do not see how differing friction could matter here. Varying depth and the associated reduction in tidal wave celerity is a much more intuitive explanation here.

**Author's response:**

Indeed, Chézy is the same for both channels, but the friction force is not because it depends on flow velocity as well. Due to the difference in depth the importance of the friction will be also

different in both channels resulting in the different results. That is what we mean in the previous version. We will clarify this in the new version.

**Author's changes in manuscript:**

We clarified this reasoning in Line 262-264 in the revised version (Line 265-267 in the marked up version) as follows:

"Because the depth in the channels influences the tidal dynamics (by for example the relative importance of friction and by difference in tidal propagation speed due to the different initial depths), the tide-induced flows were different at the junction and stayed different during the entire simulation."

5. **Comment from referee:**

L289 Processes instead of reasons? L326 Findings L351 asymmetrically L354 "relatively ratio" a word is missing here.

**Author's response and changes in manuscript:**

We modified them according to this comment in Line 324, Line 382, Line 407, and Line 409, respectively (Line 328, 389, 415, 417 in the marked up version).

6. **Comment from referee:**

Figure 2-7. The size and colormaps make these figures very difficult to gather information from. I recommend either adding 2-4 contours to the plots or just using 2-4 colors instead of a spectrum. Basically all of the detail in plots like Figures like Fig7a can't be seen by the reader.

**Author's response:**

We changed the colormap into 4 colour contour which is more clearly for the reader to see the deepening and shallowing

**Author's changes in manuscript:**

The colormap of Figure 3-7 was changed with four colour contour in Line 542, 548, 553, 557, and 560 in the revised manuscript (Line 553, 560, 566, 571, 575 in the marked up version). The example of improved Figure 4 (the time-stack of one of the scenario (depth difference 1)) is shown below.

[Figure]

Figure C: Same plot as Figure 3 but for simulations of Depth1. The panels from top to bottom show the results from different simulation (Depth1_Q2800, Depth1_Q1596, Depth1_Q500, respectively) while from left to right show the upstream channel, shallow branch, and deep branch, respectively. Different scale of colour bar for different channel is applied.

**Referee 3 (spesific comments)**

We thank the Referee for his time to carefully read the manuscript and for his comments that helped us improving the manuscript.

1. **Comment from referee:**

   1) I think the authors should take more in consideration some of the recent theoretical

   results.

   The analytical model of Redolfi et al. (2016), which represents a fully two-dimensional extension of the Bolla Pittaluga et al. (2003, 2015) model, has been proven to effectively predict the stability of both gravel and sand bed bifurcation (Redolfi et al. 2019) with the key advantage of avoiding the calibration of a specific parameter (like the α parameter of Bolla Pittaluga el al., 2003). I think these recent advancements should be at least mentioned in the Introduction and/or in the Discussion (Section 4.2).

   Similarly, in the work of Salter et al. (2017) which specifically focuses on the effect of downstream conditions on the bifurcation stability, should be considered in the Introduction.

   Please ignore this comment if you think it is too personal (I am the main author of the above-mentioned papers).

   **Author's response:**

   We agree that the mentioned work is a good extension of the work of Bolla Pittaluga et al (2003; 2015), although the fundamentals are still the same. Furthermore, we are also aware of the work of Salter et al. (2017).

   **Author's changes in manuscript:**

   We added these references in both introduction (Line 57-63) and discussion (Line 394) in the revised version of the manuscript (Line 59-65 and 401-402 in the marked up version).  We did not fully compare out model results with the mentioned papers because by using a 2DH model we don't need a nodal point relation.

2. **Comment from referee:**

   2) Line 52: this sentence is misleading, as Bolla Pittaluga et al. (2003) did not test the effect of meandering bends. 3) Line 97: the sentence "the 2D approach also results in reliable morphodynamic simulations" is very vague, as the reliability of depth-averaged models clearly depends on the specific problem under consideration.

   **Author's response:**

   Line 52: We agree that the text is not very clear in the previous version. We meant that the transverse channel slope, as studied by Kleinhans et al. (2008), was similar as in Bolla Pittaluga et al. (2003).  However, Kleinhans et al. (2008) added the effect of meandering upstream channel in the transverse transport calculation.

   Line 97: Since in this study we are interested to investigate a large scale morphological development, simulating a detail 3D result is not our goal. Therefore, 2D model is sufficient and can reduce the computational cost.

**Author's changes in manuscript:**

We removed the citation to Bolla Pittaluga et al. (2003) for this sentence to avoid misunderstanding (sentence in Line 53-54 in the revised version of the manuscript (Line 55-56 in the marked up version)).

In line 101-105 in the revised version (Line 103-107 in the marked up version), we strengthened and elaborated our argument why we chose 2D model instead of 3D according to the reasoning that we mention in the Author response.

3. **Comment from referee:**

4) Section 4.2: it would be useful to provide an indication of the average Shields number.

**Author's response:**

We agree. We will provide a table of Shields numbers and flow conditions at the bifurcation.

**Author's changes in manuscript:**

We provided a table that provides the information of: tide averaged Shields number, maximum Shields number, mean flow velocity, and $M_2$ velocity amplitude at the bifurcation in Line 606-609 in the revised version (Line 624-627 in the marked up version).

4. **Comment from referee:**

5) Code availability: to enable the reproducibility of the results, I recommend the authors to share the configuration files.

**Author's response:**

We will provide the configuration files for all simulations in the supplement.

**Author's changes in manuscript:**

The supplement was mentioned in Line 412-413 in the revised version (Line 420-421 in the marked up version).

5. **Comment from referee:**

6) System of coordinates: I cannot see the reason for placing the bifurcation at x=220 km an y~9 km. What is the meaning of the origin (i.e. x=0, y=0) point? Setting x=0 and y=0 at the bifurcation node would have been more meaningful, and it would have facilitated the reading of the maps.

**Author's response:**

This coordinate system was due to the way the grid was built in which x=0 is located at the upstream end of upstream channel.

**Author's changes in manuscript:**

We moved the origin of the x and y coordinate system to the bifurcation, e.g. in Line 532 and 538 in the revised version (Line 541 and 548 in the marked up version).

6. **Comment from referee:**

7) Units: space between value and unit is sometimes missing (e.g., line 177); please also note that units should not appear in italic (Equation 1);

**Author's response and changes in manuscript:**

The space was added in the new version for all missing space (Line 161, 201, 278, 279, 306) (Line 164, 200, 281,282, 310 in the marked up version). Equation 1 was also fixed (Line 136).

7. **Comment from referee:**

8) Graphics: in several figures (e.g., Figures 7 and 11) the labels are disproportionately small; in Figure A1: the quantities represented on the two axes are "x" and "y", not "x-axis" and "y-axis.

**Author's response and changes in manuscript:**

The labels and the axes were improved according to this comment in the new version (e.g. Line 542, 548, 581, 585, 592, 598).

[revised manuscript text omitted]
_0.25 | 0.23 | 0.20 | 0.25 | 0.54 | 0.58 | 0.59 | 0.33 | 0.42 | 0.35 | 0.51 | 0.49 | 0.71 |

630